# The olfactory gating of visual preferences to human skin and visible spectra in mosquitoes

Diego Alonso San Alberto[1,4], Claire Rusch[1,4], Yinpeng Zhan[2], Andrew D. Straw[3], Craig Montell[2] & Jeffrey A. Riffell [1✉]

Mosquitoes track odors, locate hosts, and find mates visually. The color of a food resource, such as a flower or warm-blooded host, can be dominated by long wavelengths of the visible light spectrum (green to red for humans) and is likely important for object recognition and localization. However, little is known about the hues that attract mosquitoes or how odor affects mosquito visual search behaviors. We use a real-time 3D tracking system and wind tunnel that allows careful control of the olfactory and visual environment to quantify the behavior of more than 1.3 million mosquito trajectories. We find that $CO_2$ induces a strong attraction to specific spectral bands, including those that humans perceive as cyan, orange, and red. Sensitivity to orange and red correlates with mosquitoes' strong attraction to the color spectrum of human skin, which is dominated by these wavelengths. The attraction is eliminated by filtering the orange and red bands from the skin color spectrum and by introducing mutations targeting specific long-wavelength opsins or $CO_2$ detection. Collectively, our results show that odor is critical for mosquitoes' wavelength preferences and that the mosquito visual system is a promising target for inhibiting their attraction to human hosts.

[1] Department of Biology, University of Washington, Seattle, WA 98195, USA. [2] University of California, Santa Barbara, Santa Barbara, CA 93106, USA. [3] Institute of Biology I & Bernstein Center Freibug, Albert-Ludwigs-Univesität Freiburg, Freiburg im Breisgau, Germany. [4] These authors contributed equally: Diego Alonso San Alberto, Claire Rusch. ✉email: jriffell@uw.edu

The behavioral preference of insects for certain bands in the visible light spectrum plays a profound role in structuring ecological communities by mediating processes such as plant-insect/predator-prey interactions and disease transmission[1–3]. For biting insects, such as mosquitoes, tsetse flies, and kissing bugs, vision plays an essential role in various behaviors, including flight control, object tracking for host- or nectar-finding, and locating oviposition sites[4]. The visual stimuli that mediate these behaviors are integrally tied to other host-related cues, such as scent and heat. For instance, when combined with an odor lure, tsetse flies are highly attracted to what humans perceive as blue color[5,6], and kissing bugs prefer visual objects only when also associated with odors[7]. Visually guided mosquito behaviors are also thought to play a role in host attraction[8–10]. It has long been known that mosquitoes are attracted to dark, high-contrast objects[9,11], which has led to the development of black traps[10]. For high-contrast visual stimuli, recent work has shown that certain odors stimulate visual search behaviors in *Aedes aegypti* mosquitoes. This species is not attracted to black objects in the absence of $CO_2$, but after encountering a $CO_2$ plume, they become highly attracted to such objects[11]. Other cues (heat, water vapor, skin volatiles) mediate behaviors such as landing and biting[11–13].

Despite the potential importance of color in mediating mosquito biting behaviors, surprisingly, details regarding other wavelengths that attract mosquitoes or how odors sensitize that attraction remain unclear. The visual spectra of important resources can be diverse and dominated by short and medium wavelengths (e.g., flowers or oviposition sites) or long wavelengths (e.g., human skin) (Fig. 1a, b). Despite interest in developing traps and lures that exploit mosquito spectral preferences, only a few studies have examined these preferences, and the results of those studies have been contradictory. For instance, studies of *Ae. aegypti* have shown no difference in spectral preference in the 450–600 nm wavelength range[14,15]. By contrast, other studies have demonstrated specific preferences but for different wavelength bands: *Ae. aegypti* mosquitoes were attracted to blue in one study[16] and only to green–yellow in other studies[17,18]. Other studies have shown that mosquitoes sometimes prefer red[14,17,19], although it is thought that mosquitoes lack opsin receptors sensitive to these wavelengths. Because many mosquito species are attracted to dark visual objects, responses to long wavelengths (red to human observers) may represent achromatic responses from visual channels that are sensitive to medium-wavelengths and therefore are perceived as dark gray or black when presented against a light-colored background. Nevertheless, these prior studies did not characterize the actual flight trajectories of the mosquitoes, nor control for the change in behavioral state associated with the smell of a host. Accurate control of both a visual object's reflectance and its contrast with the background is required to determine whether mosquitoes are attracted to specific wavelengths.

Here in this study, we use a large wind tunnel and a computer vision system to close knowledge gaps regarding mosquito visual and olfactory responses by examining *Ae. aegypti* free-flight responses to objects of different wavelengths, with and without the presence of $CO_2$. *Aedes aegypti* provides an excellent model for studies aimed at elucidating spectral preferences and determining how these preferences are modulated by odor. *Aedes aegypti*, which are active during the dawn and dusk periods[20], have 10 rhodopsins, 5 of which are expressed in the adult eye[21]. Little is known about opsin tuning, although they are orthologs of medium-wavelength sensitive opsins (green), and previous electroretinogram (ERG) studies suggested that *Ae. aegypti* is sensitive to medium-long wavelengths in the green–yellow spectrum[22,23]. We show that when encountering odor, mosquitoes become particularly attracted to hues that are dominant in human skin. We also demonstrate

that knockout of either the olfactory channel that gates visual attraction or the opsins that allow detection of objects that reflect long wavelengths eliminates attraction to skin tones.

## Results

**Olfactory gating of spectral preferences of *Ae. aegypti* mosquitoes.** Examining olfactory and visual search behaviors in mosquitoes often requires simulating conditions in which the statistics of the stimuli (e.g., intensity, duration) and resulting mosquito behavior are as natural as possible. We therefore examined *Ae. aegypti* behavior in a large wind tunnel spanning 450 mosquito body lengths and equipped with a 16-camera, real-time tracking system for monitoring and quantifying mosquito behaviors[11,24]. A checkerboard pattern was projected on the bottom (floor) of the wind tunnel, and a low-contrast gray horizon was projected on each side of the tunnel to provide optic flow (Fig. 1c). Similar to our previous assays, we placed two identically sized circles (3 cm diameter) on the floor of the tunnel in the upwind area of the working section, 18 cm apart and 33 cm from the odor source (Fig. 1c). In each experimental trial, 50 mated *Ae. aegypti* females were co-released into the tunnel, and their trajectories were recorded over a 3 h period (1.3 million total trajectories were recorded, with an average trajectory duration of 3 s). Simultaneous release of the mosquitoes provided an efficient method to examine their olfactory-visual responses, and was not statistically different from when the mosquitoes were released singly (see Materials and Methods—Statistical analyses for details). The tunnel was filled with filtered air for 1 h, after which a $CO_2$ plume (95% filtered air, 5% $CO_2$) located 33 cm away and separate from the visual objects was introduced into the tunnel and left for 1 h (Fig. 1d). Measurements of the plume in the working section showed an exponential decay, typical of turbulent diffusion, with a concentration of ~1500 ppm ~30 cm from the odor source. In the last hour of the experiment, only filtered air was released into the wind tunnel.

During exposure to filtered air, the mosquitoes exhibited random behavior to the odor source and visual objects, and they spent much of their time exploring the ceiling and walls of the tunnel and rarely investigated the visual objects (Fig. 1e). By contrast, upon exposure to the $CO_2$ plume, the number of flying mosquitoes more than doubled (Fig. 1f, h; Wilcoxon signed-rank test, number of trajectories in Air vs. $CO_2$, $P < 0.002$). During this time, the mosquitoes exhibited odor-tracking behavior, spending most of the time in the working section's central area with significantly elevated flight velocities (Fig. 1f, s1; Kruskal–Wallis test: $df = 2$, Chi-square = 597.23, $P < 0.0001$). The $CO_2$ also triggered an attraction to visual objects. Here, we define attraction as the amount of time a trajectory spends around an object relative to the evenly reflecting control (white to the human observer). The *Ae. aegypti* mosquitoes showed no interest in the objects during the filtered air treatment (only 1–4% of mosquitoes investigated), but during $CO_2$ release, the percentage and number of mosquitoes investigating the visual objects increased significantly (21%; paired Student's $t$ test: $P = 0.002$) (Supplementary Fig. S1). After the plume was stopped, the attraction to the visual objects ceased (Fig. 1e, Supplementary Fig. S1e; Wilcoxon signed-rank test, $CO_2$ vs. Post-$CO_2$: $P < 0.001$).

To ensure that *Ae. aegypti* mosquito visual preference behaviors were in response to discrete wavelength bands rather than object contrast, we employed visual stimuli in the range 430–660 nm (violet to red to a human observer), with each visual stimulus having the same approximate contrast with the background (Fig. 1g). While investigating the visual objects, the mosquitoes would fly upwind and hover immediately downwind of a visual object, at ~3–5 cm, while exhibiting brief excursions

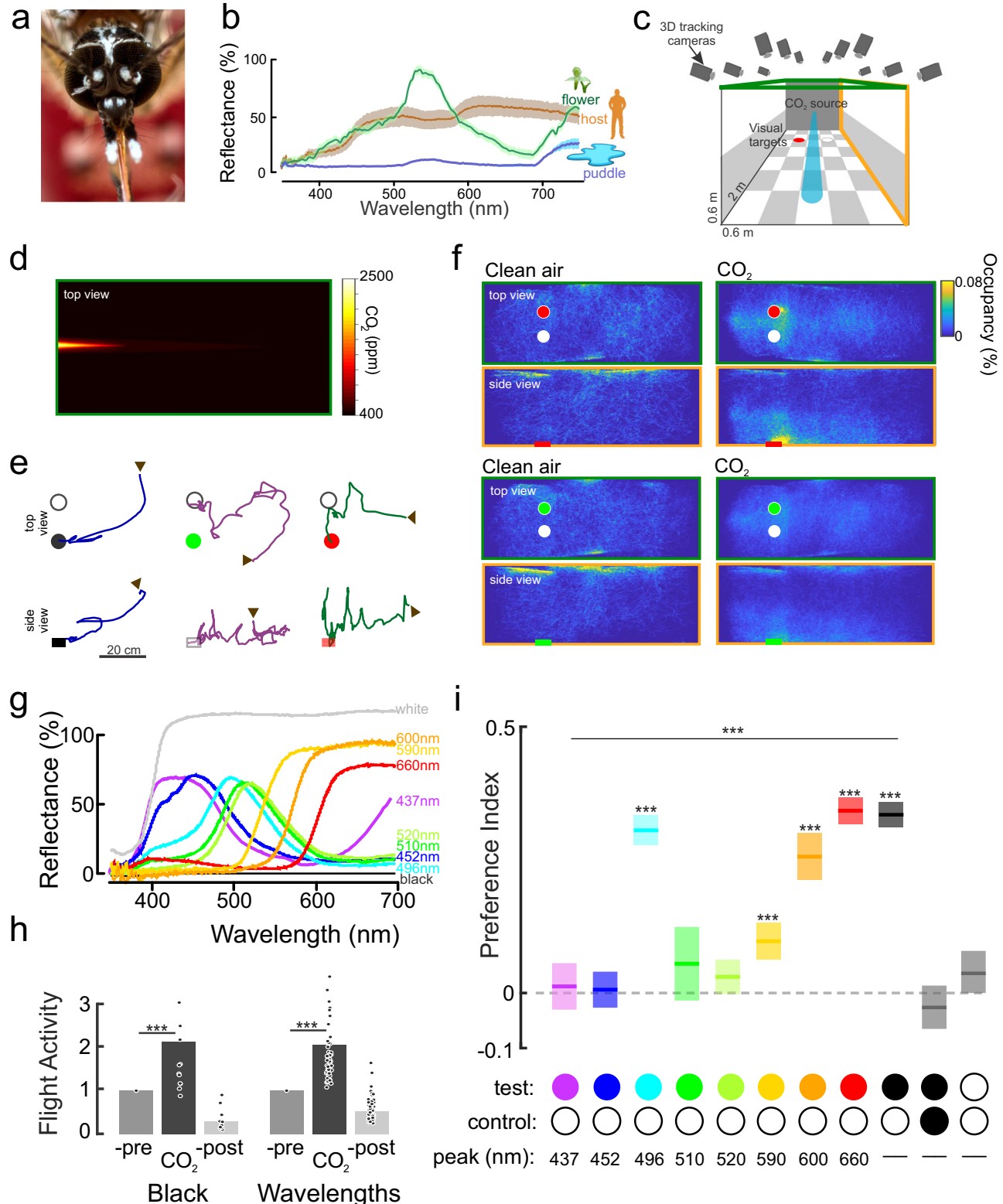

before returning to the objects (Fig. 1e). During CO$_2$ release, the same total number of mosquitoes was recruited to the visual objects of differing dominant wavelengths (Supplementary Fig. S1e). Relative to the evenly reflecting object (white to the human observer) that served as a control, however, the mosquitoes preferred certain dominant wavelengths, such as 496 nm and longer wavelengths ≥590 nm as demonstrated by their focused clustering around the object (Fig. 1e). By contrast,

other dominant wavelengths (437 nm, 452 nm, 510 nm, and 520 nm; appearing violet, blue, green and green–yellow to the human observer, respectively) elicited no attraction responses compared to the evenly reflecting control (Fig. 1e, i). Across all dominant wavelengths, CO$_2$ had a strong effect on flight velocity and duration, but there were no significant differences between treatments (Supplementary Fig. S1b, d; Kruskal–Wallis test: Chi-square < 16.82, $P > 0.11$), demonstrating that the presence of CO$_2$

**Fig. 1 Olfactory gating of mosquito color preference. a** The *Ae. aegypti* eye. Image courtesy of Alex Wild (with permission). **b** Spectral reflectance of behaviorally important objects for *Ae. aegypti* females: human skin (brown line); flower (*Platanthera obtusata*; green line) and within a small puddle filled with *Ae. aegypti* larvae (blue line). Lines are mean; shaded area around the mean is the ±sem (*n* = 6–10). **c** Wind tunnel system with real-time tracking system, odor and visual stimulation. **d** Heat map of the $CO_2$ plume in the wind tunnel. **e** Example of individual trajectories (top: [*x*-,*y*-axes], bottom: side view [*x*-,*z*-axes]). The arrows represent the start of a trajectory; circles are the visual objects. **f** Heat (occupancy) maps showing the distribution of female mosquitoes without (left panels) and with $CO_2$ delivery (right panels) while in presence of a white and a red objects (top), or white and green objects (bottom). Plots shows the top and side views of the tunnel working section. **g** Reflectance of the visual stimuli used in the experiments, as quantified using a spectrophotometer and calibrated with a white Spectralon standard. Different colored traces correspond to the different stimuli. **h** Relative flight activity between the different phases of the experiments (pre-, $CO_2$ and post-$CO_2$) for black and white circles (*n* = 12) and color and white circles (*n* = 51). There were no significant differences when the tested visual object was black vs. objects of different wavelengths (Kruskal–Wallis test, df = 1, Chi-sq = 0.01, *P* = 0.92), although for both groups $CO_2$ significantly elevated the number of flying mosquitoes compared to the filtered Air treatment (*P* < 0.002). **i** Mean preference index for the test object vs. the control (white) object. There was a significant effect of color on the attraction to the tested object (Kruskal–Wallis test, Preference index ~ pair of visual stimuli used: df = 8, Chi-sq = 597.23, *P* < 0.0001). Several hues were significantly more attractive than the control, white object (one-sample two-tailed *t*-test: ***: *P* < 0.001). Boxplots are the mean (line) with 95% confidence interval (shaded area) (*n* = 25,529; 17,729; 53,786; 23,694; 34,343; 31,037; 32,257; 24,774; 42,595; 20,929; and 48,198 mosquito trajectories for the all-white, all-black, black-and-white, Bv-T2, Bw-, Gw-T1, Gc-, YGc-, Yw-, O- and R-Hue treatments, respectively).

is necessary for attraction to specific bands of the visual spectrum, and that object attraction did not result from higher flight velocity increasing the probability of a mosquito randomly encountering a visual object.

To further investigate *Ae. aegypti* mosquito spectral preferences, a preference index value (defined as the time spent investigating a spectral object minus the time investigating the evenly reflecting control, divided by the sum of the times spent investigating both objects) was calculated for each mosquito that investigated a visual object. The dominant wavelengths differed significantly in terms of mosquito preference (Fig. 1i, Kruskal–Wallis test: df = 8, Chi-square = 597.23, *P* < 0.0001); several dominant wavelengths were more attractive to mosquitoes than the evenly reflecting object that served as a non-attracting control (Fig. 1i, one-sample *t*-test: *P* < 0.001). As the dominant wavelengths transitioned from 510 to 660 nm, the attractiveness of the object also increased. For instance, 600 nm and 660 nm visual objects were strongly preferred by female mosquitoes (multiple comparison Kruskal–Wallis test: *P* < 0.05), whereas 510 nm, 452 nm, and 437 nm objects (green, blue and violet to the human observer, respectively) were not more attractive than the control object (multiple comparisons Kruskal–Wallis test: *P* > 0.05). However, mosquitoes were not strictly attracted to the longest wavelengths, as they were also significantly attracted to objects with a dominant wavelength at 496 nm (cyan to the human observer). As a control to test for the effect of visual object attraction relative to other regions of the working section, we examined the preference between the evenly reflecting (control) object and a randomly selected volume in the wind tunnel. Compared with the randomly selected volume, female mosquitoes investigated the evenly reflecting control object significantly more in presence of $CO_2$ (preference index = −0.53 ± 0.03 (mean ± sem), one-sample *t*-test: *P* < 0.001).

**Behavioral preferences for orange-red wavelengths reflected from human skin.** Across all skin tones and differences in pigmentation, human skin is dominated in the long-wavelength range (590–660 nm) (Fig. 2a)[25], but it is unclear which bands in the human skin spectrum are most attractive to *Ae. aegypti* mosquitoes. To examine whether *Ae. aegypti* exhibit different preferences for certain spectral bands reflected by human skin, we first utilized color cards designed for cosmetics purposes to match human skin tones (Pantone SkinTone Guide) (Fig. 2a). Behavioral experiments were performed in the wind tunnel to individually test various faux skin tones that ranged from light to dark (Y02, Y10, R10, and an unpleasant orange shade typical of

individuals using cheap tanning lotion [which we designated "vile 45"]) using the evenly reflecting object (white to the human observer) as a control. Similar to the previous experiments, only during $CO_2$ release did mosquitoes become highly attracted to skin tones (Fig. 2c, e; Kruskal–Wallis test: df = 8, Chi-square = 184.37, *P* < 0.001), exhibiting no behavior characteristic of attraction before exposure to $CO_2$ (paired *t*-test: *P* = 0.09). Moreover, the mosquitoes exhibited the same level of attraction to each skin tone, with the skin tones being not significantly different from one another (Fig. 2c, e; Kruskal–Wallis test with multiple comparisons: *P* > 0.05).

To determine which region of the human skin visual spectrum is most attractive to *Ae. aegypti*, we overlaid the R10 skin color card with optical filters to attenuate discrete bands (Fig. 2b). Whereas the 450 nm optical filter had no significant effect on behavioral attraction to the skin tone compared with the positive controls (Fig. 2f; Kruskal–Wallis test with multiple comparisons: *P* > 0.58), filters blocking longer wavelengths (550–700 nm) reduced the attractiveness of the visual object (*P* < 0.05). In particular, application of the 600 nm filter was associated with a 300% reduction in attraction compared with the positive controls (Fig. 2c–f). Importantly, results for controls consisting of an overlaid infrared (IR) filter or a clear nylon coverslip did not differ significantly from the unmanipulated skin tone (Kruskal–Wallis test with multiple comparisons: *P* > 0.33). To further examine the importance of visual and olfactory integration in controlling *Ae. aegypti* visual preferences, we examined single and double mutants of the long-wavelength photoreceptors *opsin-1* and *opsin-2* and a line with a mutation in the Gr3 receptor[14], which transduces $CO_2$ signals (Fig. 2g). Whereas the *Gr3*-heterozygote, *opsin-1* and *opsin-2* single mutants, and wild-type control mosquitoes were significantly attracted to the skin tone during $CO_2$ exposure (one-sample t-test: *P* < 0.001), neither the *opsin-1,opsin-2* double mutant nor Gr3 mutant were attracted to the skin tones (Fig. 2h; one-sample t-test: *P* > 0.31).

The behavioral preference to the long wavelengths in the skin color cards may not reflect mosquito behaviors to the hues reflected from human skin. To examine this further, we tested *Ae. aegypti* mosquitoes in a smaller opaque cage (45 cm × 30 cm × 30 cm) where they were exposed to two small windows (16 cm²) (Fig. 3a, b). The cage and its environment were constructed to prevent uncontrolled contamination from thermal or olfactory cues from the human volunteer, and the windows being made from clear, heat absorptive glass to block the radiant heat from the skin (Supplementary Fig. S2). Subsequent testing showed no contamination during the experimental trials. The back of a hand was displayed in one window, and the back of a heat-protective white

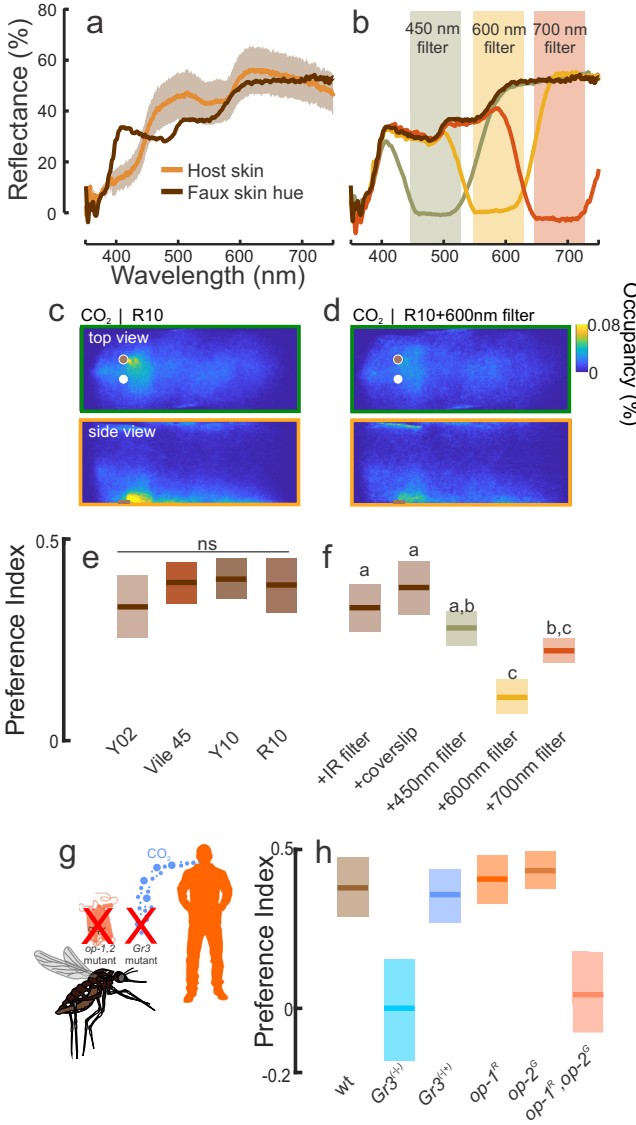

**Fig. 2 The contribution of orange-red wavelengths in attraction to faux human skin. a** Spectral reflectance of human skin and faux skin used in behavioral experiments. Lines are the mean and shaded area is the ±sem ($n = 6$). **b** Ultra-thin optical filters (450 nm, 600 nm, and 700 nm) attenuated discrete bands in the object's reflected spectrum. **c**, **d** Occupancy maps of the mosquito's distribution around the visual objects during exposure to $CO_2$. During $CO_2$, mosquitoes were significantly attracted to the faux skin color compared to the white (control) object (**c**). However, the optical filter attenuating the 550–630 nm band reduced the number of mosquitoes investigating the faux skin color (**d**). **e**, **f** Mosquitoes significantly preferred the faux human skin colors (**e**), although optical filters in the yellow to red wavelengths significantly decreased the attractiveness of the visual object (**f**). Boxplots are the mean (line) with 95% confidence interval (shaded area); letters denote statistically significant differences between groups. **g** Mosquito lines deficient in long-wavelength opsins (*opsin1* and *opsin2*), or unable to detect $CO_2$ (*Gr3* mutant), were tested in their attraction of human skin color. **h** Mean preference indices for the Gr3 mutants (blue) and opsin mutants (orange). All mosquito lines showed similar preferences to the white and skin color visual objects during exposure to filtered air (Kruskal–Wallis test: df = 3, Chi-sq = 1.68, $P = 0.64$). However, during $CO_2$ the lines were significantly different from one another in their visual preferences (Kruskal–Wallis test with multiple comparisons: df = 3, Chi-sq = 96.01, $P < 0.001$): only the heterozygote ($Gr3^{-/+}$) and wild-type (LVP) lines showed significant attraction to the skin color (one-sample *t*-test: $P < 0.001$), whereas the opsin double mutant line (*op-1/op-2*) and the $Gr3^{-/-}$ mutants showed no attraction (one-sample two-tailed *t*-test: $P > 0.31$). Boxplots are the mean (line) with 95% confidence interval (shaded area) ($n = 13{,}999$; 15,035; 14,016; 48,624; 19,284; 20,713; 40,690; 26,135; 39,649; 16,208; 3,550; 9,277; 13,799; 10,948; and 5,679 mosquito trajectories for the Y02, Vile 45, Y10, R10, IR filter, coverslip, 450 nm filter, 600 nm filter, 700 nm filter, *wt*, $Gr3^{-/-}$, $Gr3^{-/+}$, *op1-R*, *op-2G*, and *op1-R,op2-G* treatments, respectively).

glove was displayed in the other window (as an internal control). Similar to the assays in the wind tunnel, we found that mosquitoes were highly activated by $CO_2$ (Fig. 3c–e), and this increased their visual attraction to visual stimuli, including skin (Fig. 3b, c, f; Kruskal–Wallis test: df = 2, Chi-square = 84.04, $P < 0.001$). Mosquitoes showed no preference during control experiments with two white gloves displayed in the window, but significantly preferred skin (Fig. 3f; Kruskal–Wallis test with multiple comparisons: $P < 0.001$). However, when optical filters were placed over the window, blocking the longer wavelengths (550–700 nm), the attraction was significantly reduced (Kruskal–Wallis test with multiple comparisons: $P < 0.001$) and not significantly different from the negative control (Fig. 3f; Kruskal–Wallis test with multiple comparisons: $P = 0.34$). Collectively, these results demonstrate that the long-wavelength band of the visual spectrum plays an important role in determining mosquito attraction to skin color. In addition, knockout of either visual or olfactory detection receptors suppresses mosquito visual attraction to long-wavelength host cues.

**Spectral sensitivity of the *Aedes* eye.** The preference of *Ae. aegypti* for long wavelengths in the orange-red band motivated us to examine the sensitivity of the retina by recording ERGs that extracellularly measure the summed responses of retinal cells to

visual stimuli (Fig. 4a). In the first series of experiments, a moving bar of differing dominant wavelengths (blue [peak 451 nm], green [537 nm], or red [640 nm], all at the same intensity; 18° wide at 30°/s clockwise) was projected on a black background while conducting the ERG recordings (Fig. 4b). When the moving bar reached the mosquito's visual field, the ERG exhibited a negative response that quickly returned to baseline after the bar moved past the mosquito's field of view (Fig. 4b). A significant difference in wavelength-evoked responses was observed (Kruskal–Wallis test: df = 3, 40.03, $P < 0.001$), with the 451 nm and 537 nm bars eliciting the strongest responses (Fig. 4b). Although responses to the 640 nm bar were significantly weaker, this dominant wavelength still elicited ERG responses that were significantly higher than the baseline and those of the no-stimulus controls (Wilcoxon signed-rank test: $P < 0.001$).

To further characterize *Ae. aegypti* spectral sensitivity, we used a scanning monochromator to examine ERG responses across the near ultraviolet (UV) to far-red wavelength range (350–750 nm). *Ae. aegypti* exhibited the highest sensitivity to short (410 nm, 3.2 mV) and medium wavelengths (520 nm, 2.8 mV) (Fig. 4c, d). Strong ERG responses (0.47–1.27 mV) were still noted in the medium-long wavelengths (>590 nm), although at >700 nm, the responses decreased to approximately 0.25 mV, which was still significantly higher than the baseline control (Wilcoxon signed-rank test, $P < 0.03$).

**The role of visual contrast in determining *Ae. aegypti* preferences.** Mosquitoes are very sensitive to detecting dark objects that contrast highly with the background[8,11]. In the above experiments, we kept the total object contrast (400–700 nm) with the background approximately the same, but the *Ae. aegypti* preference for long-wavelength objects and skin tones motivated

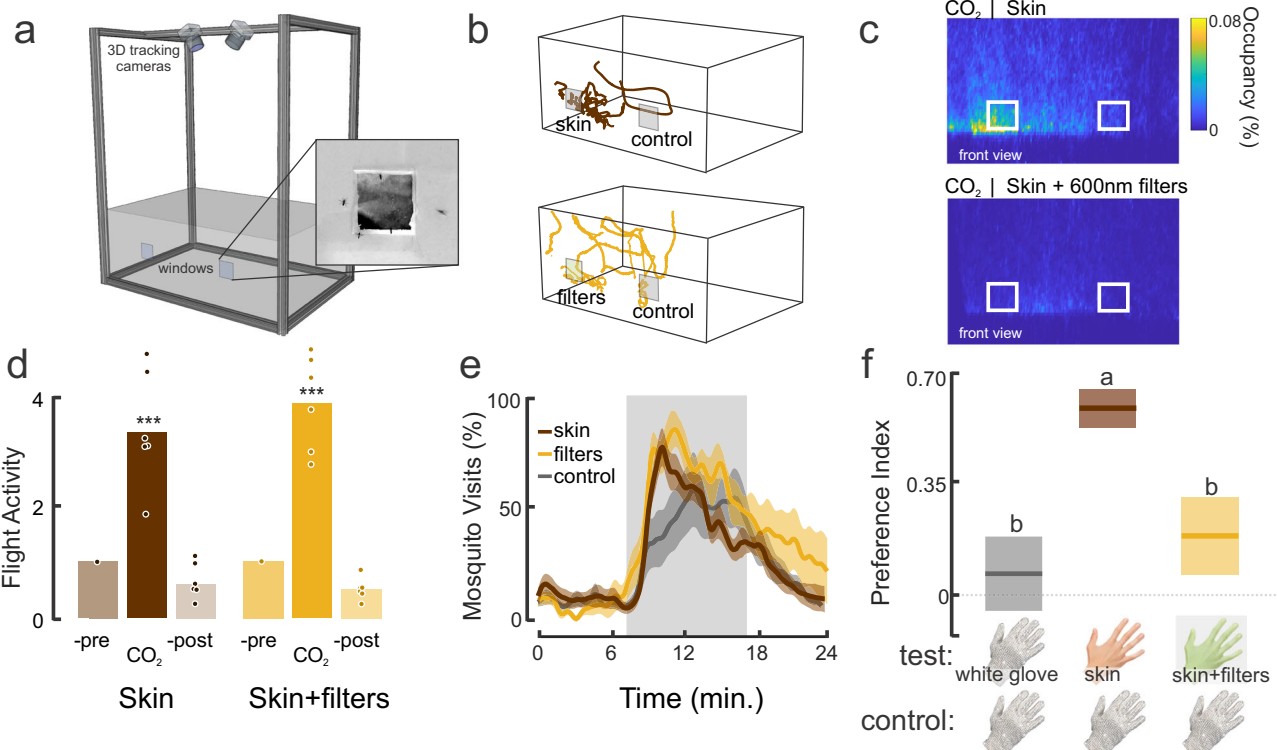

**Fig. 3 The importance of long wavelengths in attraction to human skin. a** Cage assay with real-time tracking system, odor, and visual stimulation through two windows on the front of the cage. **b** Example of individual trajectories (top: skin and control (white glove), bottom: skin+filters (550–700 nm), and control). **c** Occupancy maps showing the distribution of female mosquitoes during $CO_2$ stimulation while in presence of the skin and control (top), and the skin+filters (550–700 nm) and control (bottom). **d** Relative flight activity between the different phases of the experiments (pre-, $CO_2$, and post-$CO_2$) for the skin and skin+filters treatments ($n = 6$ trials/treatment). There was no significant difference in the relative activity during the $CO_2$ phase between the skin and skin+600-nm filter treatments (Kruskal–Wallis test, df = 1, Chi-sq = 0.004, $P = 0.96$). **e** The percentage of mosquitoes visiting the windows over the duration of the experiment. Few mosquitoes investigated the windows before the $CO_2$ exposure. However, exposure to $CO_2$ significantly increased the numbers of mosquitoes visiting the windows relative to the pre-$CO_2$ period (Kruskal–Wallis test with multiple comparisons: df = 5, Chi-sq. = 277.85, $P < 0.0001$), although during $CO_2$ there were no significant differences in the total number of mosquitoes investigating the windows between treatment groups (Kruskal–Wallis test with multiple comparisons: $P > 0.57$). Lines are the means and shaded areas the ±sem. **f** Mean preference index for the different treatment groups (white glove vs. white glove, skin vs. white glove, and skin + filter (550–700 nm) vs. white globe). Boxplots area the mean (line) with 95% confidence interval (shaded area). Different letters denote statistically significant differences between groups (Kruskal–Wallis test with multiple comparisons, $P < 0.01$). ($n = 13,597$ for the skin treatment group; $n = 9502$ for the for the skin + filters treatment group; and $n = 9368$ for the control group).

us to evaluate whether these responses were due to contrast alone (calculated as the Weber contrast, or the difference in spectral energy reflected by an object and the background, divided by the sum of the two) or whether mosquitoes can discriminate long-wavelength objects >590 nm independently of intensity. As the first step in determining how features of a visual stimulus impact mosquito visual preference, gray objects that contrasted differently with the background were tested against the evenly reflecting control (Fig. 5, Weber Contrasts: −0.28–0.02). Similar to the above results and across all tested stimuli, the presence of $CO_2$ increased mosquito flight activity and the number of visits to the visual objects (Fig. 5c, Wilcoxon signed-rank test, Air vs. $CO_2$: $P < 0.001$). Female mosquitoes exhibited significantly greater attraction to the majority of the gray visual objects than the control object (Fig. 5d, one-sample $t$-test: $P < 0.001$). However, when exposed to the lightest gray object, which closely approximated the background and the evenly reflecting control, mosquitoes showed no preference for either object (Fig. 5d; Student's $t$ test: $P = 0.33$). Overall, object darkness and contrast with the lighter background was significantly related to mosquito preference, with mosquitoes investigating and preferring darker objects (Fig. 5d; Kruskal–Wallis test: df = 5, Chi-square = 634.16,

$P < 0.001$). Although *Ae. aegypti* showed a distinct preference for darker objects, mosquito flight velocity and duration did not significantly differ across treatments (Kruskal–Wallis test: Chi-square > 7.40, df = 5, $P > 0.055$).

**Effects of contrast and spectral discrimination on *Aedes* preferences.** Given the difference between high retinal sensitivity to medium-long (green) wavelengths, lack of attraction to those objects, and the relatively low ERG sensitivity to long (orange/red) wavelengths but strong behavioral attraction to those objects, we asked the following question: what role does the darkness of the object (i.e., its contrast) have on behavioral preferences vs. the object's dominant wavelength? Although we lacked the photo-receptor tunings that would allow us to normalize the inputs between spectral channels, we experimentally manipulated object contrast to equalize the perceptual contrast of the objects to the mosquitoes and then examined the mosquito's ability to discriminate between different bands of the visual spectrum.

We first performed behavioral experiments in which we altered the darkness of the gray objects to determine the contrast that matched the attraction shown to the 660 nm object (red to human

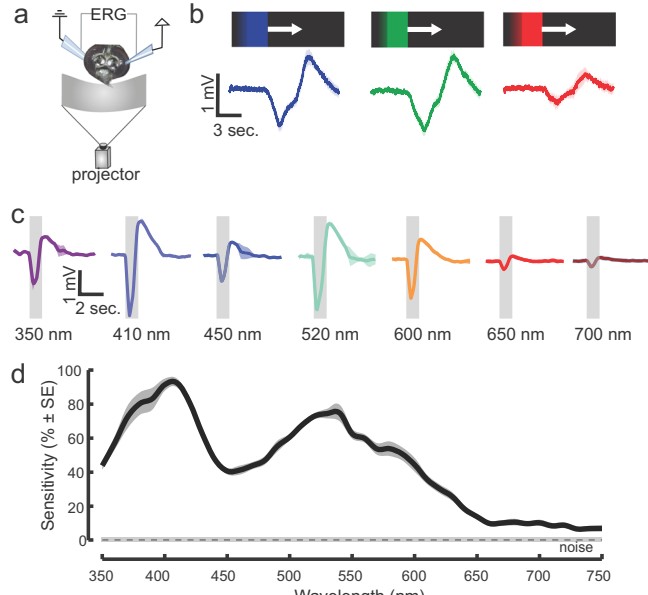

**Fig. 4 Retinal sensitivity to visual stimuli. a** Experimental setup for the electroretinogram (ERG) experiments. **b** ERG responses to a blue (410 nm), green (520 nm) or red (>590 nm) moving bar (mean ± sem; $n = 7$ mosquitoes). Responses to the moving object were significantly higher than the baseline for all tested dominant wavelengths (410 nm, 520 nm or 590 nm), although 410 nm and 520 nm bars elicited stronger responses (Kruskal–Wallis test, Amplitude responses–wavelength moving bar, df = 2, Chi-sq. = 40.03, $P < 0.001$, $n = 7$ mosquitoes). **c** ERG responses to pulses of light (350–750 nm). Traces are the mean responses (shaded area is the ±sem; $n = 8$ mosquitoes) to discrete wavelengths showing the elevated responses to 410 nm (violet to human observer) and 520 nm bands. **d** Retinal sensitivity curve across the tested wavelengths from 350 to 750 nm. Two maxima occurred at the short (420 nm) and medium (~530 nm) wavelengths, although responses were still significantly elevated above the noise at wavelengths more than 650 nm (two-sided $t$-test, ERG vs. noise: $P < 0.01$).

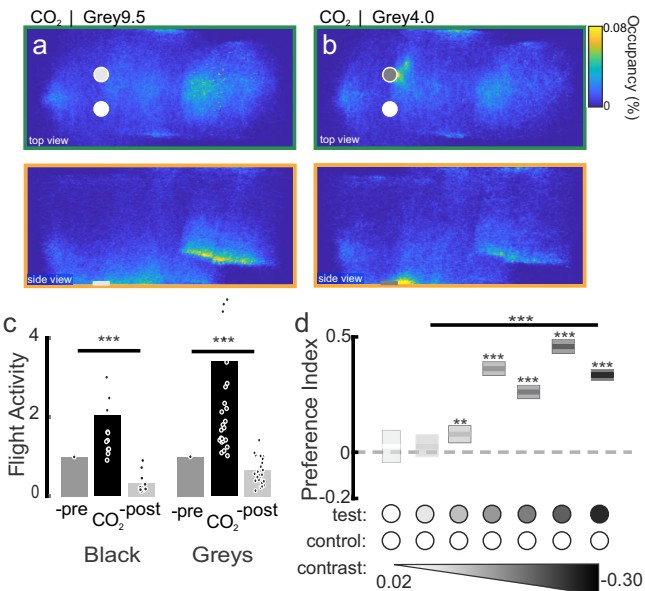

**Fig. 5 The effect of achromatic contrast on mosquito attraction to visual objects. a** Occupancy map of the distribution of female mosquitoes in the wind tunnel (top and side views) during $CO_2$ delivery in presence of a white and a light gray object (Weber Contrast: −0.05). **b** As in a, except the gray object has a higher contrast (−0.23). **c** Relative flight activity between the different phases of the experiments (pre-, $CO_2$ and post-$CO_2$). Mosquitoes exhibited similar flight activities across all tested visual objects (Kruskal–Wallis test, df = 1, Chi-sq = 3.24, $P = 0.07$). **d** Mean preference indices for the test (gray, or black) vs. control object (white) with 95% confidence interval ($n = 12{,}764$; 27,537; 37,085; 28,644; 36,050; 25,896; and 21,514 mosquito trajectories for the white, grey9.5, grey6.5, grey4.5, grey4.0, grey2.5, and black treatments, respectively). Object contrast had a significant effect on the attraction to the tested object (Kruskal–Wallis test: Preference Index~contrast tested object, df = 5, Chi-sq = 634.16, $P < 0.001$). All gray objects were significantly more attractive than the control, white object (one-sample tw-tailed $t$-test, **: $P < 0.01$, ***: $P < 0.001$) except for the lightest gray object (−0.05), which was not more attractive (one-sample $t$-test: $P = 0.33$).

observer)(Fig. 6a, b). Across a range of gray contrast levels, the 660 nm object was significantly more attractive to *Ae. aegypti* than the gray objects (Fig. 6a, c, e; Kruskal–Wallis test: Chi-square = 149.75, df = 6, $P < 0.0001$). As the gray objects became darker, however, they became more attractive to the mosquitoes, and the strong preference for the 660 nm object decreased (Fig. 6e). The attractiveness of the 660 nm object equaled that of the darkest gray and black objects (Fig. 6e; one-sample $t$-test: $P = 0.07$ and $P = 0.08$ for the darkest gray [gray 1.5] and black objects, respectively). We then determined which contrast of a 510 nm object (green to human observer) matched the attraction to the 660 nm object. For this purpose, we used the darkest gray object that was equally attractive as the 660 nm object and tested it against objects with dominant wavelengths of 510 nm but differing in their background contrast (Fig. 6b). The dark gray object was significantly more attractive than most 510 nm objects, but the darkest 510 nm object elicited the same level of attraction (Fig. 6f; preference index = −0.01; one-sample $t$-test: $P = 0.66$).

To determine whether *Ae. aegypti* mosquitoes can discriminate between objects of different dominant wavelengths but similar levels of apparent contrast, we tested attraction to the 660 nm vs. the darkest 510 nm object (Fig. 6g, h). During exposure to $CO_2$, mosquitoes were strongly attracted to 660 nm but not the dark 510 nm object (Fig. 6g, h; one-tailed $t$-test: $P < 0.0001$). We also determined whether mosquitoes could discriminate between

objects that overlapped in their spectral bands. For this purpose, we matched the apparent contrast of the non-attractive 452 nm object (blue to human observer) to an attractive gray. Next, we tested mosquito responses to the 497 nm object (another attractive spectral band) and the dark 452 nm object, as well as the 497 nm object vs. the dark 510 nm object. Similar to the results observed with the 660 nm object, mosquitoes significantly preferred the 497 nm object over the dark 452 and 510 nm objects (Fig. 6h; one-tailed $t$-test: $P < 0.0001$). Thus, mosquitoes easily discriminated between objects with overlapping and distinct spectral bands, even when the object contrasts matched.

By incorporating object contrasts, reflectance values, and peak wavelengths as independent variables into a series of linear models, the results of the behavioral tests offered a means to examine the relative contributions of these variables toward mosquito preferences. In these models, all possible combinations were tested, and the best model was selected based on its Akaike Information Criterion (AIC) score, where the AIC estimates the value of each model and lower scores reflect the quality of the statistical model. Using combinations of the independent variables, we found the best model (and hence, lowest AIC score) relied on object peak wavelength and contrast, and excluded reflectance (Supplementary Fig. S3a, b). But which dominant wavelengths might be critical for mediating these behaviors? To further explore the relationship between the object

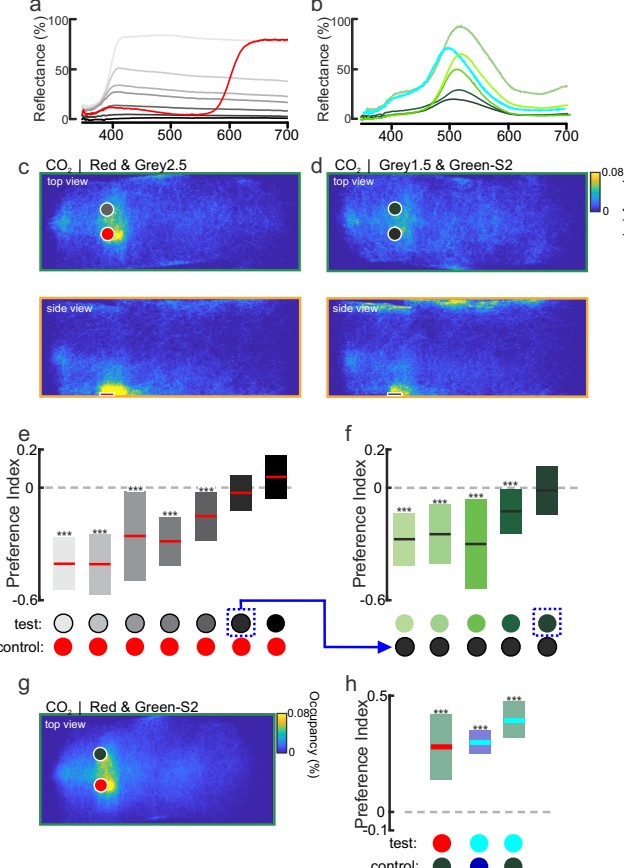

wavelengths and its attractiveness, a series of linear models were run using a multivariate analysis (Principal Components Analysis, PCA) of each object's visual spectrum. The PCA analysis allowed reduction of highly collinear and dimensional spectral data into reduced components, which can then be used as independent variables in the model. The best model explained ~17% of the variance in object attractiveness (Supplementary Fig. S3c) and indicated that preference is negatively correlated with the content of the medium wavelength band (500–575 nm) in the object's visual spectrum (Supplementary Fig. S3d, e).

**Comparison of wavelength preference between mosquito species.** The strong and specific responses of *Ae. aegypti* to specific bands in the visual spectrum and the similarity in long-wavelength opsin gene expansion in other mosquito species[21] motivated us to examine the wavelength preferences in *Anopheles* (*An.*) *stephensi* and *Culex* (*Cx.*) *quinquefasciatus* mosquitoes. To examine the visual preferences in *An. stephensi* and *Cx. quinquefasciatus*, we first conducted ERG recordings of 7-day-old females and examined their retinal responses to discrete wavelengths from 350 to 750 nm (Fig. 7a). Both mosquito species exhibited the strongest response to short wavelengths (350–420 nm; UV-visible violet to the human observer), and the second strongest response was observed in the medium wavelengths (500–520 nm; cyan-green). The strong response of *An. stephensi* and *Cx. quinquefasciatus* to the short wavelengths of 370 nm contrasted with that of *Ae. aegypti* (dashed line), which exhibited the strongest response at ~400 nm (Figs. 3d and 7b, c). All three species exhibited similar responses to the long wavelengths in the orange to red band (620–750 nm).

Are mosquito species' behavioral preferences for visual objects correlated with their ERG responses, and are their color preferences similar? To answer these questions, we tested the responses of *An. stephensi* and *Cx. quinquefasciatus* mosquitoes to objects that dominate in wavelengths at 452 nm, 510 nm, 660 nm, and black objects in the wind tunnel using a methodology similar to that used for *Ae. aegypti*, except at lower light levels (1.28 μW/cm²). As with the results for *Ae. aegypti*, encountering the $CO_2$ plume caused a doubling in the number of flying mosquitoes and increased the percentage of mosquitoes that investigated the visual objects 5.58- to 9.15-fold relative to air-only treatment for *An. stephensi* and *Cx. quinquefasciatus*, respectively (Fig. 7d; Kruskal–Wallis test: df = 2, Chi-square > 7, P < 0.01). However, in contrast to the tight clustering around attractive visual objects by *Ae. aegypti* (Fig. 1f), occupancy maps showed that the responses of *An. stephensi* and *Cx. quinquefasciatus* mosquitoes were much more diffuse (Fig. 7e, f). Nonetheless, *Cx. quinquefasciatus* formed a clustering hotspot around the outlet of the odor plume nozzle that was much stronger than the responses of the other two species (Figs. 1f, 7e, f). Although we tried to minimize the odor nozzle's visual signature by using clear acrylic and tubing, *Cx. quinquefasciatus* mosquitoes might have located the plume source based on the high $CO_2$ concentration or by seeing the nozzle to some degree.

We next examined the preferences of the mosquito species for different spectral objects (black, blue, green, or red) relative to the evenly reflecting white object (the non-attractive control). Similar to *Ae. aegypti*, when both *An. stephensi* and *Cx. quinquefasciatus* mosquitoes were subjected to filtered air treatment, they showed no preference for the black object or any of the spectral objects relative to the white control object (preference indices of -0.04 and 0.08 for *An. stephensi* and *Cx. quinquefasciatus*, respectively; Kruskal–Wallis test: df = 3, Chi-square < 3.66, P > 0.30). How-ever, their spectral preferences changed when exposed to $CO_2$. After encountering the $CO_2$ plume, *An. stephensi* preferred the

**Fig. 6 The role of dominant wavelength vs. contrast in mosquito visual attraction. a** Spectral reflectance of 660 nm (R-Hue), black, and gray objects used in the experiments: Gray 9.5, 6.5, 4.5, 4.0, 2.5, and 1.5, with Weber contrast values of −0.17, −0.30, −0.05, −0.10, −0.20, −0.24, −0.27, and −0.28, respectively. **b** Spectral reflectance of 496 nm and 510 nm objects used in the experiments: Gw-T1, Gw-T3, Gc-T1, Gc-Hue, G-S1, and G-S2, with Weber contrast values of −0.18, −0.11, −0.17, −0.20, −0.25, and −0.27, respectively. **c, d** Occupancy maps showing the distribution of female mosquitoes in the wind tunnel (top and side views) during $CO_2$ delivery in presence of the 660 nm (R-Hue) and Grey2.5 objects (**c**), or 510 nm (G-S2) and Grey1.5 objects (**d**); both the 510 nm and Grey1.5 objects have the same levels of contrast with the background (Weber Contrasts of −0.27). **e** Mean preference indices for the 660 nm vs. gray objects with different levels of contrast with the background. The Grey1.5 object (Weber Contrast value of −0.28) was not significantly different from 0 (one-sample two-tailed t-test: P = 0.07) and was subsequently used in experiments in (**f**) (blue arrow). **f** The Grey1.5 object was significantly more attractive than most of the 510 nm objects (one-sample two-tailed t-test: P < 0.001, ***), although the darkest 510 nm object (G-S2; Weber Contrast = −0.27) was not significantly different from 0 (one-sample two-tailed t-test: P = 0.66)) (n = 10,098–15,578 trajectories for each tested object). **g** As in (**c**), occupancy maps showing the distribution of trajectories around the 660 nm and dark 510 nm objects. **h** Mean preference indices for 660 nm vs. dark 510 nm objects, and 496 nm vs. dark 452 nm or dark 510 nm objects. Mosquitoes significantly preferred the 660 nm and 496 nm objects over the dark chromatic objects (510 nm and 452 nm)(one-sample two-tailed t-test: P < 0.0001. For (**e, f** and **h**), boxplots area the mean (line) with 95% confidence interval (shaded area), and asterisks denote P < 0.001 (one-sample t-test) (n = 7191–27,717 mosquito trajectories for each tested object).

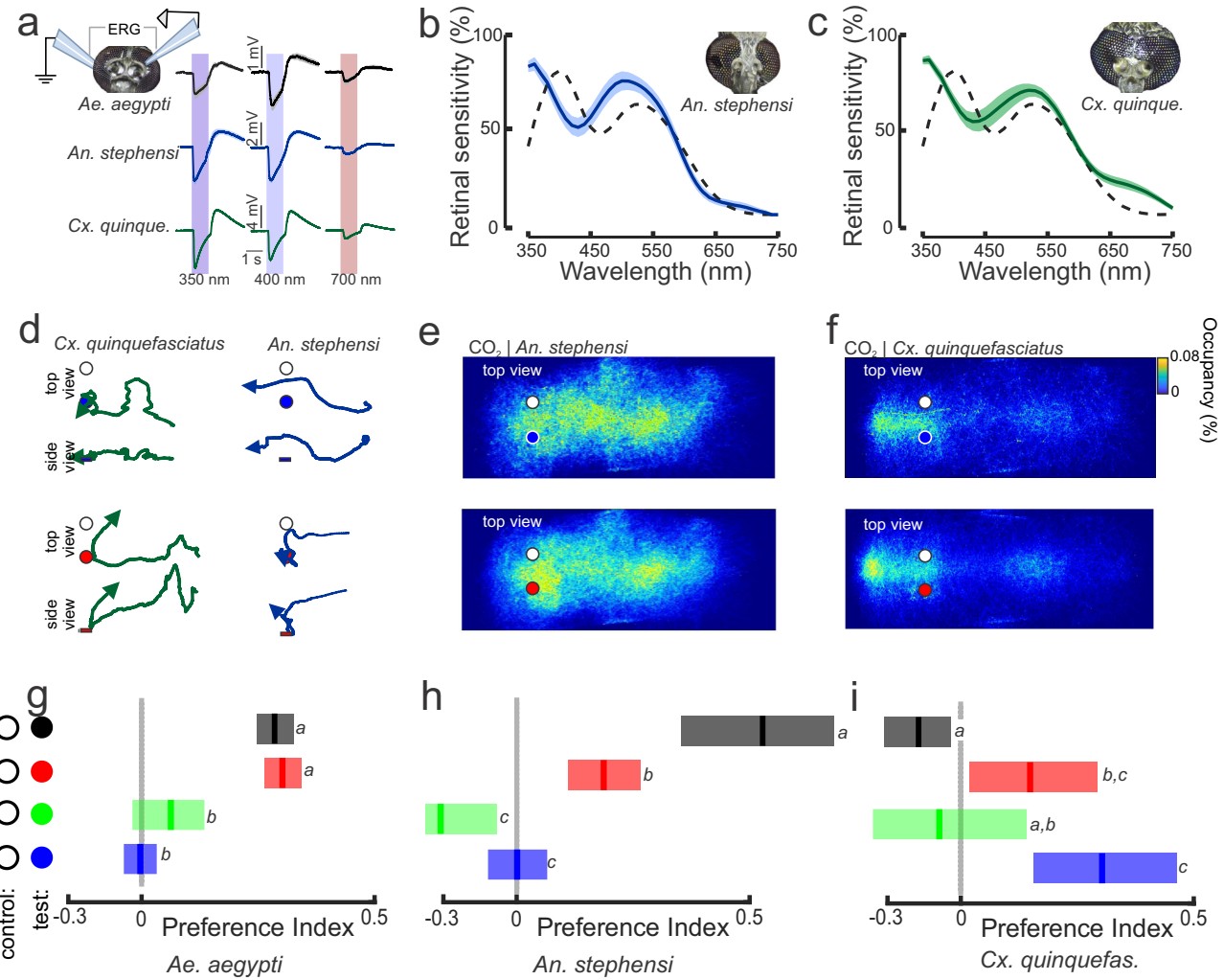

**Fig. 7 Species-specific responses to dominant wavelengths. a** Electroretinogram (ERG) recordings from *Ae. aegypti*, *An. stephensi* and *Cx. quinquefasciatus* mosquitoes. Each trace is the mean ± sem of 7–9 mosquitoes/species. **b, c** Retinal sensitivity to discrete wavelengths show that both *An. stephensi* (**b**) and *Cx. quinquefasctiatus* (**c**) have the strongest responses in the UV (360 nm) and green (520 nm) bands. The dashed line is the retinal sensitivity of *Ae. aegypti*. **d** Representative flight trajectories [(x,y) and (x,z)] of *Cx. quinquefasciatus* (left) and *An. stephensi* (right) mosquitoes. *Cx. quinquefasciatus* showed a mild attraction to the blue object (top), whereas *An. stephensi* showed an attraction to the red object (bottom). **e, f** Occupancy maps (x,y) of *An. stephensi* (**e**) and *Cx. quinquefasciatus* (**f**) distribution around the 452 nm (top) and 660 nm (bottom) objects during $CO_2$ exposure. **g–i** The preference indices for *Ae. aegypti* (**g**), *An. stephensi* (**h**) and *Cx. quinquefasciatus* (**i**) in response to the black, 660 nm (R-Hue), 510 nm (Gc-T1), and 452 nm (Bw-T1) objects. Lines are the means and shaded bars are the 95% confidence intervals, and letters above bars denote statistical comparisons (Kruskal–Wallis test with multiple comparisons: $P < 0.05$) (for *Ae. aegypti*, $n = 29,254$; 13,580; 18,190; and 12,086 trajectories; for *An. stephensi*, $n = 5817$; 9134; 2153; and 5627 trajectories; and for *Cx. quinquefasciatus*, $n = 3746$; 13,553; 4238; and 2835 for black, 660 nm, 510 nm, and 452 nm treatments, respectively).

black and 660 nm objects (Fig. 7h; Kruskal–Wallis test: df = 3, Chi-square = 38.6, $P < 0.001$), but they were not significantly attracted to the 452 nm or 510 nm objects (Fig. 7h; multiple comparison Kruskal–Wallis test: $P > 0.05$). By contrast, *Cx. quinquefasciatus* mosquitoes preferred the 452 nm and 660 nm objects (Kruskal–Wallis test: df = 3, Chi-square = 13.6, $P < 0.01$; with multiple comparisons: $P < 0.05$) but were not significantly attracted to the 510 nm or black objects (Fig. 7i; multiple comparison Kruskal–Wallis test: $P > 0.05$). Collectively, these results show that odor strongly sensitizes attraction to visual objects across mosquito species; however, spectral preferences can be species-specific.

## Discussion

Free-flight behavioral experiments with *Ae. aegypti* mosquitoes have shown that these insects integrate olfactory, visual, skin volatiles, and thermal cues to function efficiently and robustly in

complex environments[11,13,26]. However, we know very little about mosquito visual-guided behaviors or how vision is involved in host selection. In this study, we utilized real-time tracking of mosquito behaviors in a large wind tunnel. The wind tunnel system enables control of aerodynamic conditions to structure the odor plume and allow the plume and visual objects to be decoupled in time and space. Both are important considerations when testing olfactory-visual integration. In this study, and similar to our previous work using achromatic objects[11,26], the presence of $CO_2$ increased mosquito responses to visual objects in a wavelength-specific manner. Both chromaticity and contrast were important components in visual object attraction and could partly explain *Ae. aegypti* mosquito preferences for objects that appear orange and red to human observers. These results were qualitatively similar to those reported by Smart and Brown (1957), who examined the landing responses of mosquitoes on colored cloth in a field[16]. The results of their study showed that

mosquitoes (*Aedes* sp.) are attracted to red and black cloths, with darker colors being more attractive than lighter shades[16]. Our ERG results and wind tunnel assay data suggest that mosquitoes can detect and are attracted to the long-wavelength bands in the orange and red portions of the human visual spectrum, although if objects contrast highly with the background and/or are darker, then they become more attractive. The demonstrated spectral preferences of mosquitoes and their lack of attraction to spectral bands in the 450 nm region, and 510–550 nm regions (blue and green to human observer), impacted the fashion industry in the early twentieth century. For instance, Nuttall and Shipley (1902)[27] suggested wearing khaki pants would be appropriate for ensembles in a tropical environment, and the US military changed its dress shirts from dark blue to light blue in part to mitigate mosquito biting[28]. Nonetheless, a key question is what drives these responses in mosquitoes, particularly the attraction to darker and higher-contrast colors? An important component could be the visual OFF responses to the visual background[29]. In our wind tunnel, we projected a light gray checkerboard pattern on the bottom of the tunnel to provide optic flow and contrast with the visual objects. The lack of opsins for the long wavelengths could cause an OFF response in downstream neurons that receive input from the photoreceptors. One complication with this hypothesis is that mosquitoes still preferred the long wavelengths when tested against a dark 510 nm object that matched it in apparent contrast, suggesting they have an ability to detect these long wavelengths. This was further demonstrated by the ERG responses to long wavelengths. Recent work in *Drosophila* has shown that photopigments provide a mechanism for flies to long wavelengths >600 nm[30], and it is possible that similar processes play a role in mosquitoes.

Although the contrast with a lighter background did impact the attraction responses of *Ae. aegypti* mosquitoes, the preferences for the dominant wavelengths that appear red and cyan to humans were greater than the apparent contrasts of the competing visual cues (objects with dominant wavelengths at 510 nm, or 452 nm). The *opsin-2* gene, which is tuned to the green to orange band of the visual spectrum, is highly expressed in the mosquito retina. The results of this study and those reported elsewhere[24] suggest that *opsin-1* and *opsin-2* play important roles in determining object preference and visual attraction to human skin. Additional research will be needed to identify the opsins and neural circuits involved in small-field object detection and color preference and determine how odor modulates those responses.

Compared with other insects, such as honeybees or the tsetse fly, we know little about mosquito visual ecology or how visual cues are integrated with other senses in these insects. Whereas shallow pools of water can be rich in medium-long wavelengths and flowers dominant in short-medium wavelengths, hosts' skin is dominated by long wavelengths in the >600 nm region of the visual spectrum (Fig. 1b). Abundant work by the cosmetics industry has shown that human skin—irrespective of skin tone or pigmentation—has a lower peak in the green wavelength (530 nm, ~20%) and a dominant reflectance in the long wavelengths (>600 nm, 20–60%). The diurnal *Ae. aegypti* and nocturnal and crepuscular *An. stephensi* and *Cx. quinquefasciatus* mosquitoes are all active during periods in which these longer wavelengths are dominant. For example, *Ae. aegypti* exhibit peak activity in the mornings and late afternoons, and *An. stephensi* and *Cx. quinquefasciatus* mosquitoes are especially active during moonlit nights—both environments are long-wavelength shifted[20,31,32]. In this study, we show that mosquitoes are especially sensitive to long wavelengths (590–660 nm) for host detection; blocking these wavelengths can suppress object attraction. Moreover, we found that mosquitoes can distinguish between overlapping and discrete spectral bands (Fig. 6h), even

when their apparent contrasts match. Insect photoreceptors within an ommatidial cartridge transduce light intensity and spectral information. At the photoreceptor terminals, they also provide antagonistic inputs to downstream neuron targets, thus allowing discrimination of spectral inputs (termed 'color opponency'). In *Drosophila melanogaster*, color opponency at the photoreceptor terminals plays an important role in their color discrimination and preference[29], and it could be that similar processes are at play in mosquitoes. In a variety of insects, including flies, bees, and butterflies, spectrally sensitive photoreceptors form connections with transmedulla neurons that project into the lobula, where additional spectral and motion processing—including color opponency—occurs[33–35].

It is important to note that our current experiments did not incorporate close-range cues from a host, such as heat, water vapor, or skin volatiles. These cues play critical roles in controlling landing and biting behaviors, and future work could determine how visual spectra are processed in tandem with these other stimuli. Nonetheless, previous work has shown that visual cues can promote mosquito orientation and search behaviors in combination with odor, heat, or water vapor[11,13]. The integration of multimodal stimuli in driving behavioral responses raises questions regarding how the sensory systems are linked in the brain. Odor stimulation increases visual responses in the object-detecting neuropil in the *Ae. aegypti* lobula[26]. Neuropil in this brain region is responsive to moving objects but not wide-field motion. Interestingly, whereas olfactory stimulation increases visual responses in the lobula, visual stimulation does not modulate glomerular responses in the antennal lobe, the primary site for processing olfactory information in the mosquito brain. Why might this occur? Mosquitoes have a relatively poor visual resolution (~10°); thus, vision may not provide fine-scale information about the identity of an object. Instead, an object's odor may provide information about its identity, whereas vision can provide details regarding the location of the object.

Despite the potential importance, few studies have examined retinal responses to long wavelengths in mosquitoes[22,23], and how peripheral and downstream visual circuits, such as those in the optic lobes, process this information remains unknown. Evolutionary analyses of long-wavelength opsins in diverse mosquito species have suggested these genes are functionally important. In *Ae. aegypti*, *Anopheles coluzzii*, and *Cx. quinquefasciatus*, these genes have undergone duplication events and may be under positive selection, explaining the commonalities in ERG responses. *Aedes aegypti* has 10 putative opsins, five of which are potential medium- to long-wavelength opsins in the adult (>500 nm)[21]. One of these genes, *opsin-1*, is expressed in the largest group of photoreceptor cells (R1–R6) in the *Ae. aegypti* eye[22]. The R1-R6 photoreceptors form a trapezoid-like structure in the ommatidial cartridge, with the R7 and R8 cells in the middle. Perhaps similar to *D. melanogaster* with rhodopsin-1 (Rh1), the *Ae. aegypti* opsin-1 is likely involved in motion- and dim-light sensitivity[36]. The inner R7 and R8 photoreceptors are involved in color vision, and the *Ae. aegypti* opsin-2 is expressed in the R7 photoreceptors located in distinct bands on the dorsal and ventral surfaces of the eye, and possibly used for navigation and biting behaviors. For mosquitoes, opsins that differ in spectral tuning can be co-expressed in the R7 photoreceptor in the female retina (e.g., the long-wavelength *opsin-2* and short-wavelength *opsin-9*), thereby increasing their range in wavelength sensitivity[37]. This may contrast the opsin co-expression in *D. melanogaster* R7 cells in the dorsal region of the eye, where two short (UV)-sensitive opsins are co-expressed[38], presumably to increase the spectral contrast with the green-absorbing opsins in the R8 layer. The opsin co-expression and broadening of spectral input in mosquitoes could be advantageous under low-light

conditions, as photon capture would be maximized, thus allowing for detection of suitable hosts or perhaps sources of nectar[37,39], but this advantage would come at the cost of detection of particular wavelength bands. In our study, however, mosquitoes were able to discriminate between distinct and overlapping spectral bands (510 nm and 660 nm; 496 nm and 510 nm; and 496 and 452 nm), all at the same apparent contrasts, suggesting that other opsins, perhaps in the R8 cells, or downstream circuits, play a role in increasing the separability of those hues.

Preferences for certain dominant wavelengths plays a critical role in a diverse array of insect vectors, and it is integrally tied to olfaction. For instance, traps that incorporate visual and olfactory cues have proven transformative as a low-cost method for controlling tsetse flies in parts of Africa[40]. Like *Ae. aegypti*, the majority of tsetse flies (e.g., *Glossina morsitans morsitans* and *G. pallidipes*) locate hosts based on smell, and once they are within close range (<10 m), visual cues cause the insects to investigate and potentially bite if the object is a host. Beginning with Vale (1974)[41] and continuing into the 1990s, researchers found that flies are most attracted to a specific blue (~460 nm), followed by red and black, but they are not attracted to green and white[5,6]. Researchers subsequently found that incorporating blue and black hues in traps was particularly effective at inducing flies to come into contact with insecticide-treated screens. *Triatoma infestans* (i.e., the kissing bug) is another example of an insect that integrates odor and visual cues to mediate attraction to targets. Exposure to an aggregation pheromone modified the responses of *T. infestans* to colored objects, including the red, blue, and black hues, although the bugs always reject green (~525 nm) and white hues[7]. In a similar manner, our results obtained from tests of different mosquito species demonstrate the importance of olfaction in mediating mosquito spectral preferences. In the absence of $CO_2$, mosquitoes did not demonstrate any preference between evenly reflecting (white to humans) and dominant wavelengths, but they became attracted to certain dominant wavelengths in the presence of $CO_2$. However, there were species-specific differences in their wavelength preferences. Whereas *Ae. aegypti* was equally attracted to both 660 nm and black objects, *An. stephensi* was most attracted to black, followed by 660 nm objects. By contrast, *Cx. quinquefasciatus* was attracted to 452 nm, followed by 660 nm objects; surprisingly, however, this species was not attracted to small black objects. Collectively, the results of our current study and those of other studies show that the visual systems of insect disease vectors and their behaviors constitute attractive targets for the development of traps incorporating visual features that can be species-specific in terms of attraction, thus providing incentive to identify molecular targets that compromise mosquito olfactory-visual responses.

## Methods

**Mosquitoes, odor delivery, and wind tunnel**. Mosquitoes (*Aedes aegypti*: Rockefeller, Liverpool, *Gr3*[ECFP][13] and *opsin-1*, *opsin-2* mutant lines; *Anopheles stephensi* (Indian strain) and *Culex quinquefasciatus*) were raised at the University of Washington campus. Mosquito lines were provided from BEI Resources (Manassas, VA, USA) (*Ae. aegypti*: Rockefeller, Liverpool, *Gr3*[ECFP][13]; *An. stephensi* and *Cx. quinquefasciatus*). In the case of the *Ae. aegypti opsin-1*, *opsin-2*, and *opsin-1,opsin-2* lines, we used existing mosquito lines that were generated as previously described[24]. Briefly, the op1 and op2 alleles were generated by selecting short-guide RNAs that targeted the GPROp1 (LOC5568060) and GPROp2 (LOC5567680) loci[24]. Lines were homogenized and verified by PCR before testing[24] (Supplementary Tables 1–3). Mosquitoes were raised in groups of 100 individuals and anesthetized with cold to sort males from females after cohabitating for 7 days. At this time, more than 90% of the females have been mated, as indicated by their developing embryos. For each experimental trial in the wind tunnel, we released 50 females into the wind tunnel working section 3 h prior to the mosquito's subjective sunset (the time period of peak activity for *Ae. aegypti*). Each visual stimulus was tested in 4–12 experimental trials (on average, ~5575 trajectories were quantified per trial) and 12,300 *Ae. aegypti* were flown in total across all experiments (99.4% of all mosquitoes flew in the working section, and thus their behaviors were captured by the real-time tracking system), for a total of 1,305,695 trajectories. After one hour, the 5% $CO_2$ plume (or filtered air in control experiments), was automatically released from a point source at the immediate upwind section of the tunnel and at a height of 20 cm and in the centerline of the tunnel. The $CO_2$ remained on for 1 h, before switching off for another hour of filtered air (post $CO_2$). The 1 h time periods were chosen because mosquitoes did not adapt or habituate to the $CO_2$ plume due to the diameter of the plume (~1.5 cm) and the size of the tunnel lowered the encounter probability of the mosquitoes to the plume. In addition, the 1 h periods provided a baseline of behavior before $CO_2$ release, and after encountering the $CO_2$, mosquitoes would remain activated and visually sensitized for 5–10 min—thus, the 1 h post-$CO_2$ period allowed sufficient time for mosquitoes to return to baseline.

The $CO_2$ and filtered air were automatically delivered using two mass flow controllers (MC-200SCCM-D, Alicat Scientific, Tucson, AZ) that were controlled by a Python script that allowed synchronizing odor and filtered air delivery with the trajectory behaviors. The $CO_2$ plume was quantified using a Li-Cor LI-6262 CO2/H2O analyzer (Li-Cor, Lincoln, NE) for a total of 500 locations throughout the tunnel (Fig. 1). Data yielded an exponential decay similar to a model of turbulent diffusion at the airflow (40 cm/s) and turbulent intensities (5%) of this tunnel, such that 20 cm from the source and parallel to the wind flow the plume was ~1700 ppm (Fig. 1c, d), which is in the range of the plume of human breath.

All behavioral experiments took place in a low-speed wind tunnel (ELD Inc., Lake City, MN), with a working section 224 cm long, 61 cm wide, by 61 cm high with a constant laminar flow of 40 cm/sec (Fig. 1). We used three short-throw projectors (LG PH450U, Englewood Cliffs, NJ) and rear projection screens (SpyeDark, Spye, LLC, Minneapolis, MN) to provide a low contrast checkerboard on the floor of the tunnel and gray horizons on each side of the tunnel. The intensity of ambient light from the projectors was 96 lux across the 420–670 nm range. A 3D real-time tracking system (the open-source Braid system)[11,42] was used to track the mosquitoes' trajectories. Sixteen cameras (Basler AC640gm, Exton, PA) were mounted on top of the wind tunnel and recorded mosquito trajectories at 60 frames/sec. All cameras had an opaque IR Optical Wratten Filter (Kodak 89B, Kodak, Rochester, NY) to mitigate the effect of light in the tracking. IR backlights (HK-F3528IR30-X, LedLightsWorld, Bellevue, WA) were installed below and the sides of the wind tunnel to provide constant illumination beyond the visual sensitivity of the mosquitoes. The temperature within the wind tunnel, measured using ibuttons and FLIR cameras (FLIR One Pro, FLIR Systems Inc., Goleta, CA USA), was 22.5 °C and did not show any variability within the working section[11,24]. Ambient $CO_2$ was constantly measured outside of the tunnel and was ~400 ppm.

**Visual stimuli and experimental series in the wind tunnel**. To determine the role of odor in the innate spectral preferences of mosquitoes, and identify the role of achromatic contrast and wavelength discrimination, a series of different experiments were conducted. In each experiment, two visual stimuli, separated by 18 cm, were presented to the mosquitoes on the floor in the upwind area of the tunnel and perpendicular to the direction of airflow. Visual stimuli consisted of paper circles that were 3 cm diameter (Color-aid Corp., Hudson Fall, NY, USA). Reflectance spectra for all the visual stimuli were characterized using an Ocean Optics USB2000 spectrophotometer with a deuterium tungsten halogen light source (DH2000) calibrated with a white Spectralon standard (Labsphere, North Sutton, NH, USA). The projector and working section light intensities were measured using a cosine-corrected spectrophotometer (HR- + 2000, Ocean Optics, Dunedin, FL, USA) 5 cm from the projector source (63 μW/cm²). The achromatic contrasts of the visual objects relative to the background were measured using the calibrated spectrophotometer and the Weber contrasts were calculated by the intensity (μW/cm²) of the object ($I_{object}$) and background ($I_{backgroun}$), where ($I_{object}$ - $I_{background}$)/$I_{background}$. In the first experimental series, *wt* (ROCK) *Ae. aegypti*, we examined the innate preference of individual colors relative to the non-attractive white object. Tested objects had distinct and peak wavelengths of 437 nm, 452 nm, 496 nm, 510 nm, 520 nm, 590 nm, 600 nm, and 660 nm (Bv-T2, Bw-, Gw-T1, Gc-, YGc-, Yw-, O- and R-Hue; Color-aid Corp.). These stimuli all had similar achromatic contrasts (−0.12 to −0.18) and peak reflectance values, but had distinct peak wavelengths (Fig. 1g). The position of the respective hue and white object in each replicate trial was randomized.

In the second experimental series, experiments were performed to examine which spectral bands of the human skin might attract mosquitoes. To ensure the replicability of the experiments, and enable the control of visual object humidity, temperature, and odor, we first elected to first use faux skin mimics (Pantone SkinTone Guide; Pantone LLC, Carlstadt, NJ 07072 USA). Using *wt* (ROCK) *Ae. aegypti*, we tested four different skin tones (R10, Y10, and Y02) and a skin tone that we named "vile 45", which matched the putrid orange tone from individuals who use cheap tanning lotion (PANTONE 16-1449X, Gold Flame). Similar to our previous wind tunnel experiments, each individual skin tone was paired with an evenly reflective object (white to human observer) that served as a control. To attenuate different spectral bands reflected from the skin tone, we used ultra-thin (~200 um thick) plastic filters that were placed over the object. Filters were selected to attenuate the 450–530 nm band, the 550–630 nm band, or the 650–730 nm band (36–333 [notch filter], 35–894 [long-pass filter], and 35–896 [long-pass filter], respectively; Edmund Optics Inc., Barrington, NJ USA). Reflectance measurements

of the object with the filters showed the transmission loss was <5% for bands outside of the filtered wavelengths (Fig. 2B). As a control to determine if attenuating a spectral band outside of the visual spectrum effects mosquito behavior, we used an IR filter that allows transmission of 350–750 nm band of the visual spectrum (14-547 [KG2], Edmund Optics Inc., Barrington, NJ USA). To control for the physical effect that placing the plastic filter over the object may have had on mosquito behavior, we used plastic coverslips with the same refractive index as glass (72261-22; Electron Microscopy Services, Hatfield, PA USA). Experiments were also conducted with $Gr3^{-/-}$ mutants[13] and the $opsin-1^{-/-}$,$opsin-2^{-/-}$ double mutants[24] to examine how the loss of olfactory or visual detection, respectively, impacted mosquito attraction to the skin color. As controls, we used heterozygote ($Gr3^{-/+}$), single mutants ($opsin-1^{-/-}$, $opsin-2^{-/-}$), and the wild-type (Liverpool) lines.

In the third series, the role of achromatic contrast, and not wavelength, on mosquito visual preferences were examined. Gray circles (3 cm diameter), differing in their Weber Contrast (−0.28 to −0.05) and ranging from near-black to very light gray (Grays 1.5, 2.5, 4.0, 4.5, 6.5, and 9.5; Color-aid Corp., Greyset), were run in combination with a white circle (Weber Contrast of 0.02). Similar to the above experiments, $wt$ (ROCK) $Ae.$ $aegypti$ were used in the experiments.

In the fourth experimental series, we examined whether the spectral preferences of the mosquito ($Ae.$ $aegypti$ [ROCK]) while controlling for the apparent contrast of the different hues. We normalized the perceptual contrasts of the visual objects by testing a range of grays of different contrasts with the background (Grays 1.5, 2.5, 4.0, 4.5, 6.5, and 9.5 [Weber Contrasts of −0.28 to −0.05]; Color-aid Corp., Greyset) in combination with, and against, the 660 nm object (R-Hue; Weber Contrast = −0.17). Once we identified the gray object that was as attractive as the 660 nm object, identified by the PI that not significantly from 0 ($t$-test: $P > 0.05$), we then tested that gray (Gray 1.5; Weber Contrast = −0.28) against a range of different objects that had the same peak wavelength (510 nm) but different Weber Contrasts, from light to dark (−0.11 to −0.27). Based on the 510 nm object that elicited the same level of attraction as the dark gray (PI = 0), we then tested the dark 510 nm object vs. the red object. Similar to experiments with the 510 nm objects, we tested the dark gray (Gray 1.5; Weber Contrast = −0.28) against a range of different objects that had the same peak wavelength (452 nm) but different Weber Contrasts, from light to dark (−0.13 to −0.28), followed by testing the 660 nm object vs. the dark 452 object, and the attractive cyan (495 nm) object vs. the dark 452 nm or dark 510 nm objects with the same apparent contrast.

**Visual attraction to skin in a Cage-assay**. To assay the visual attraction to the skin, acrylic cages (45 × 30 × 30 cm; McMaster-Carr; cat. # 8560K171) were constructed to allow for video recording and tracking from above. Thermal insulation and white sheeting were wrapped around the cage's exterior to prevent any heat cues from outside of the cage while providing a uniform visual environment to the interior. The sides and windows of the cage were sealed to prevent odor contamination from the experimenter from leaking into the cage. Using thermal imaging (FLIR One Pro, FLIR Systems Inc., Goleta, CA USA), solid-phase microextraction fibers (75 um CAR/PDMS SPME fiber, 57344-U; Supelco, Bellefonte PA USA) for VOC collection and subsequent analysis using Gas Chromatography with Mass Spectrometric Detection (Agilent Technologies, Palo Alto, CA, USA), and $CO_2$ measurements both inside and outside of the cage, allowed for the testing of odor and heat contamination within the cage. Results showed no odor, $CO_2$ or heat contamination (Fig. S2). A small 4 cm vent on the cage side facing away from the experimenter and underneath the hood ventilation system allowed filtered air or 5% $CO_2$ input to the cage. In contrast to the wind tunnel assays, ambient airflow was small (<10 cm/s) within the cage. To conduct the visual preference assays, two 4 × 4 cm windows, spaced 18 cm apart, were cut into the acrylic. Windows were sealed with heat absorptive glass (Schott KG2, Edmund Optics) to prevent both thermal and odor contamination into the cage. Similar to the visual stimuli used in the wind tunnel experiments, mosquitoes were tested with a uniform reflective "control" in one window (white-colored glove to human observer), and the other window displaying either human skin, or human skin through long-wavelength optical filters (550–730 nm band; 36–333 and 35–894 filters; Edmund Optics Inc., Barrington, NJ USA). Positions of the visual stimuli displayed in the windows were randomized between experimental replicates. To test for any side preference or contamination in the cage, control experiments were also conducted using two white gloves as the visual stimuli in the windows. Experiments were performed in a chamber held at ~20–22.5 °C, and the cage was situated underneath a hood ventilation system allowing air exchange. A $CO_2$ Flypad (Genesee Scientific; cat. # 59-119) was placed immediately adjacent to the vent on the side of the cage, and similar to the wind tunnel experiments, $CO_2$ was controlled by two mass flow controllers (MC-200SCCM-D, Alicat Scientific, Tucson, AZ) via a Python script that allowed synchronizing odor and filtered air delivery with the trajectory behaviors. Filtered air was released for the first 8 min of each experiment, followed by the release of 5% $CO_2$ for 8 min, before switching off for another 8 min (post-$CO_2$). Two cameras (Basler AC640gm, Exton, PA) were mounted above the cage and recorded mosquito trajectories at 100 frames/sec. IR backlights (HK-F3528IR30-X, LedLightsWorld, Bellevue, WA) were installed above the cage. Human skin reflectance measurements and assays were from three males and three female individuals on the University of Washington (Seattle, WA USA) campus (ages 25–46 years old), and volunteers were from various ethnic groups.

An ergonomic armstand set was used to position and keep the arms steady over the 24 min. period. Because $CO_2$ causes behavioral changes in the mosquitoes that can last 5–10 min after $CO_2$ exposure is stopped, the 24 min. experiment was the minimum time that allowed us to examine the mosquito responses at the different time periods (Air, $CO_2$, post-$CO_2$). Protocols were reviewed and approved by the University of Washington Institutional Review Board, and all human volunteers gave their informed consent to participate in the research. Similar to the wind tunnel experiments, we used 6–8 day-old, non-blood-fed, mated females who were sucrose deprived for 24 h but had access to water. We released 50 $Ae.$ $aegypti$ (ROCK) females into the cage for each experiment, and the assays were initiated 3 h before lights off (ZT12). Ambient $CO_2$ was constantly measured both inside and outside of the cage.

**Visual preferences in Cx. quinquefasciatus and An. stephensi**. $Anopheles$ $stephensi$ (Indian strain) and $Culex$ $quinquefasciatus$ mosquitoes were separately raised in groups of 100 individuals and anesthetized with $CO_2$ to sort males from females after cohabitating for 7 days. At the time of their subjective sunset, groups of 50 female mosquitoes were released into the working section of the wind tunnel. After 1 h of filtered air, 5% $CO_2$ was released from a point source at the upwind section of the tunnel (height of 20 cm, and in the center of the tunnel) for 1 h, after which filtered air was released from the point source. The intensity of ambient light from the projectors was ~1.3 μW/cm². The low-light intensity, relative to that used with $Ae.$ $aegypti$, was necessary to recruit females to the visual objects. At higher light intensities $An.$ $stephensi$ and $Cx.$ $quinquefasciatus$ mosquitoes responded to the $CO_2$ plume but they did not respond to the visual objects. We found that these two species began to investigate the visual objects only at light intensities <5 lux. Like our experiments with $Ae.$ $aegypti$, the temperature within the wind tunnel was ~22.5 °C.

**Trajectories analysis**. As described above, for each experimental trial we release a group of 50 mosquitoes because the working section of the wind tunnel was large enough to minimize any interactions between individuals, while allowing for the efficient capturing of behaviors to the $CO_2$ plume and visual objects. Our tracking system is unable to maintain mosquito identities for extended periods of time, but we considered individual trajectories as independent for the sake of statistical analysis. To ensure that the release of the 50 mosquitoes reflected behaviors of single mosquitoes in the working section, experiments were performed as described in Material and Methods—Visual Stimuli and Experimental Series in the Wind Tunnel but using single mosquitoes. A 660 nm and evenly reflective objects (red and white to the human observer) were used as the visual stimuli in these experiments. Fifty $Ae.$ $aegypti$ mosquitoes were individually tested and released into the working section of the tunnel and their behaviors were compared to mosquitoes that were co-released. Results of these experiments showed that all 50/50 individual mosquitoes flew in the tunnel, which compares with the 99.4% of mosquitoes that flew in the tunnel in all the co-released experiments. We also found that flight velocities, durations, and PIs of the individual mosquitoes to red and white circles were not statistically different from co-released mosquitoes (unpaired $t$-tests, $P = 0.75$, 0.93, and 0.44 for flight duration, flight velocity, and PI, respectively). During the $CO_2$ exposure, on average and for those mosquitoes that investigated the visual objects (38% of the 50 mosquitoes individually-released into the tunnel), one mosquito produced nine flight trajectories during the 1 h period (range from 1 to 37 trajectories) and investigated the visual object 3.70 times (±2.33). This was similar to the estimated number of trajectories per mosquito that investigated the objects in co-released experiments (4.76 ± 3.64). Since 38% of the mosquitoes investigated the visual objects, this percentage provided evidence that our trials were not biased from insufficient sampling. Therefore, co-release of the mosquitoes allowed efficient testing of mosquito behavior to the visual objects and did not differ from individually released mosquitoes.

Analyses were restricted to trajectories that were at least 90 frames (1.5 s) long. Only trajectories that lasted for more than 1.5 s were analyzed (average length trajectory: 3.1 s, longest trajectory: 96.4 s, total number of 1,305,695). The flight activity in the different phases of an experimental trial (pre-, $CO_2$ and post-$CO_2$) where the mosquitoes encountered filtered air or $CO_2$ were quantified by the number of trajectories recorded during one phase divided by the number of trajectories recorded in the previous phase. To examine the mosquito behaviors and preferences to the two visual stimuli in the tunnel, a fictive volume was created around the visual cues (area: 14 × 14 cm, height: 4 cm). The volume was centered over the object in the crosswind direction, and shifted slightly downwind in the wind line direction. This volume was chosen as it captures the area of primary activity of the mosquitoes. A sensitivity analysis was performed by adjusting the volume size and demonstrated that this volume best captured the mosquitoes investigating the visual objects while excluding mosquitoes transiting to other areas of the working section.

Occupancy maps were calculated by dividing the wind tunnel into 0.3 cm² squares. For each replicate experiment, the number of mosquito occurrences within each square was summed and divided by the total number of occurrences in all squares to yield a percentage of residency. We did not quantify landings on the spots due to limitations of the camera angles needed to identify landings. During the filtered "air" treatment, mosquitoes often investigated certain areas of the working section, such as the top or corners of the working section, causing hot

spots in the occupancy maps. This is typical for mosquito activity without a stimulus. By contrast, when $CO_2$ is released, these hot spots are no longer apparent, and instead, the mosquitoes investigate the visual objects or navigate to the odor source, as demonstrated by a hotspot in the central area of the working section or near the odor source. For each mosquito line, the replicate trials were pooled to create an occupancy heat map for the tested visual stimulus.

To calculate the fractions of trajectories that approached either visual object, for each trajectory we calculated a preference index by determining the amount of time a trajectory spent in each volume divided by the total time it spent in both volumes. If the trajectory spent all of its time in only one volume then it was assigned a preference index of 1 (test object) or −1 (white neutral object). Approximately 25–50% of the trajectories approached either object. From these preference indices, we calculated the global mean, and bootstrapped the 95% confidence interval of the mean through random resampling of the individual trajectories 500 times. To determine whether the mosquitoes preferred the visual objects compared to elsewhere in the tunnel, we calculated the preference index for each trajectory at each time point as the amount of time the mosquito spent in a particular $4 \times 14 \times 14$ cm volume that was randomly selected in the tunnel and compared them to the volumes containing the visual objects. Mean flight velocities were calculated from the 3D tracks of each individual trajectory. To further examine whether mosquito responses to the visual objects changed throughout the experiment, the percent of time (per each minute interval) the mosquitoes investigated the visual objects was calculated. Statistically significant groups were estimated using a Kruskal–Wallis, Mann–Whitney $U$-test with Bonferroni correction at a $P = 0.01$ level, or the one-sample $t$-test. All recorded data were analyzed using Matlab (Mathworks, 2019a release). To explore the potential impact of pseudoreplication on the statistical results, we used a multilevel analysis using the restricted maximum likelihood method and setting the experimental trial as a random, nested effect. We found that, for the visual stimuli used in the experiments, there was a significant impact of the visual spectra of the object on the preference index ($P < 0.001$) but the experimental trial did not contribute to the model results. Thus, potential for pseudoreplication due to biased sampling did not contribute to our results. The model was created using R 4.0.3 and the *lmer* function from the *lme4* package[43].

**Electroretinogram (ERG) recordings**. ERG recordings were performed by fixing 6 day-old, non-blood-fed female mosquitoes to a coverslip using Bondic glue. Mosquitoes were dark-adapted for 1 h prior to stimulation. The recording glass electrode (thin-wall glass capillaries; OD, 1.0 mm; length, 76 mm; World Precision Instruments, cat. # TW100F-3) was pulled using a micropipette puller (Sutter Instrument, p-2000), and filled with Ringer's solution (3 mM $CaCl_2$, 182 mM KCl, 46 mM NaCl, 10 mM Tris pH 7.2). The reference electrode, a sharpened tungsten wire, was placed into one compound eye in a small drop of electrode gel (Parker, cat. # 17-05), and the recording electrode was placed immediately on the surface of the contralateral eye. Two different types of visual stimuli were presented to the mosquitoes. In the first series, the mosquitoes were placed at the center of a semi-cylindrical visual arena (frosted mylar, 10 cm diameter, 10 cm high); a video projector (Acer K132 WXGA DLP LED Projector, 600 Lumens) positioned in front of the arena projected the visual stimuli. To test the response to moving objects, similar to what the mosquito might encounter in flight, we tested responses to a 19° wide bar moving from left to right (Clockwise) (Fig. 4). The mosquito was randomly tested with blue, green, and red bars (distinct peaks at 455, 547 nm, and 633 nm, 18 lux), and each colored bar was tested 10–30 times per mosquito ($n = 7$ mosquitoes).

The second stimulation method used a digital monochromator to examine responses to different wavelengths across the mosquito visual spectra (350–750 nm). Mosquitoes were exposed to a 1 s pulses of light (10 lux) from a light source (35-watt Halogen; ThorLabs) and a fiber optic scanning monochromator (MonoScan 2000, Mikropak GmbH, Ostfildern, Germany) that provided control of the transmitted wavelengths (±2 nm). Light was transmitted via optical fibers (QP600-1-SR-B X, Ocean Optics, FL 32792, USA) and through a neutral density filter (fused silica, Thorlabs Inc., 0-1 OD). Each mosquito preparation was tested to wavelengths of 350–750 nm in 10 nm increments ($n = 8$ mosquitoes/species). The visual stimuli were calibrated using a cosine-corrected spectrophotometer (HR-+2000, Ocean Optics, Dunedin, FL, USA) that was placed immediate to the recording preparation, allowing us to scale the irradiance of the tested stimuli. The light-induced responses were amplified by using an A-M Systems amplifier (10-100x; A-M Systems, 1800) and digitized using a Digidata data acquisition system (Digidata 1550B, Molecular Devices, San Jose, CA 95134). Data were visualized and analyzed using Matlab software (Mathworks).

**Linear models**. Linear models were created using R 4.0.3 and the lm function with the default option. Comparison between models were performed using the AIC function that calculates the Aikaike's Information Criterion (AIC) for each model. For the first series of models, the dataset consisted of the mean preference index per experiment, the contrast value, peak wavelength and brightness value for the tested object. For the second series of models, the dataset contained the mean preference index per experiment and the area under the curve (AUC) of the reflectance measurement for the tested object calculated with bins of 25 nm from 350 to 675 nm. A Principal Component Analysis (PCA) was applied to the AUC vector to remove collinearity in the object's spectrum.

**Reporting summary**. Further information on research design is available in the Nature Research Reporting Summary linked to this article.

## Data availability

The source data underlying Figs. 1h, 3d, and 5c are provided as a Source Data file. The wind tunnel data generated in this study have been deposited in the Dryad Data Repository at https://doi.org/10.5061/dryad.d51c5b04d. Source data are provided with this paper.

## Code availability

Software is available on https://github.com/riffelllab (https://doi.org/10.5281/zenodo.5579784) and https://github.com/strawlab/strand-braid.

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

## Acknowledgements

We are grateful for the advice and discussions with C.E. Reisenman and F. Van Breugel. Support for this project was funded by Air Force Office of Scientific Research under grants FA9550-20-1-0422 (J.A.R.); the National Institutes of Health under grants R01-AI148300 (J.A.R.), R21-AI137947 (J.A.R.); and an Endowed Professorship for Excellence in Biology (J.A.R.). C.M. was supported by EY008117 (C.M.) from the NEI, AI165575 (C.M.) from NIAID, DC016278 (C.M.) from NIDCD, and from the U.S. Army Research Office and accomplished under cooperative agreement W911NF-19-2-0026 (C.M.) for the Institute for Collaborative Biotechnologies. A.D.S. was supported by the Momentum program of the Volkswagen Foundation.

## Author contributions

D.A.S.A., C.R., and J.A.R. designed research; D.A.S.A., C.R., and J.A.R. performed wind tunnel experiments; C.R. and J.A.R. conducted electroretinogram experiments; A.D.S. assisted in visual stimulus, wind tunnel software, and experimental designs; Y.Z. and C.M. generated the opsin mutant lines. All authors wrote and edited the paper.

## Competing interests

The authors declare no competing interests.
