## [Peer Review File · Nature Communications]

The olfactory gating of visual preferences to human skin and visible spectra in mosquitoesREVIEWER COMMENTS:

Reviewer #1 (Remarks to the Author):

This manuscript reports interesting findings related to the potential host location strategies in the *Aedes aegypti* mosquitoes. The topic is quite interesting, and authors have utilized an array of clever techniques to address some fascinating questions that have long been puzzling mosquito biologists on what makes *Ae. Aegypti* mosquitoes so well adapted in locating humans. While olfactory biologist and chemists have long celebrated a blend of lactic acid and carbon dioxide (sometimes added with ammonia) as a strong bait in the wind tunnel -- and with decent success in the field studies -- the question remained as to how to improvise it. This m/s sheds light onto this topic by elegantly integrating the genetic/molecular techniques with sensory biological paradigms. This work significantly contributes to our fundamental understanding of the olfactory and visual components of the mosquito host seeking behavior esp, in *Aedes*, but also extends broadly to the *Culex* and *Anopheles*. This work once again proves that *A. aegypti* mosquitoes are 'well-behaved' organisms to study in the laboratory; authors heroically added a species each of *Culex* and *Anopheles*, and I commend authors for that.

I have a few major concerns about the methodologies that I believe authors should be able to address/explain fairly.

1- Why was it deemed necessary to test mosquitoes in groups (#50)? It appears that the integrity of the tracks cannot be assigned to an individual mosquito (authors state this as much; Line 680). It is very hard for me to conclude that the tracks represent a clear behavioral output from a 'behaving/responding' mosquito. Ideally, an individual mosquito's behavior would have provided a clearer track. I ask authors to explain and expand the reasons for testing mosquitoes in groups, esp. in the context of integrating the tracks. I am certain that the authors must have rigorously thought/debated about it. A clear explanation will be of great help to navigate the work. In the same context, a simple definition of "attraction" (as per the authors) right at the outset will be helpful

2- I am surprised at the integrity of the CO₂ plume. Since CO₂ is heavier than the air, it usually tends to fall down soon after the release from a point source. Authors mention hundreds of measurements to depict the Fig 2d. Can you expand?

3- Was it necessary to have the length of testing for 3hrs, with an hour of CO₂ stimulation? How would you account for habituation, and such effects?

Here are some specific comments:

Line 2 and onwards: Use of term 'mosquito' is very loose. Culicine and Anopheline misquotes have rather different and distinct biology, including sensory. It would be helpful if authors do not generalize.

Line 30. Reference 13 (Lacey and Carde's work) was done with *Culex*

Line 56. Consider replacing "... when experiencing..." with "while encountering..."

Line 65: What does it signify to say 450? Please explain.

Line 76: Are you suggesting ~1500 ppm is what a mosquito will experience at 30cm distance from a breathing human? (Considering we exhale ~4-5%)

Line 81. How would you arrive with that number? (double # getting activated)

Line 82. Velocity would assume an individual's distance. Can you please provide the basis for your assumption?

Line 89: Fig 1Se is not clear. Can you instead just indicate pre-, CO₂ and post-CO₂? (instead of scale 0-200mins). Also in the figure S1 legend, Line 9 from top, add "significantly INCREASES..."

For Figure 1b and throughout: Y axis needs to read "RefLectance"

Line 105: It may be interesting to have a measure of distance. If certain stimulus (i) induce more kinetic effects.

Line 157. Legends have small letters whereas figures have CAPS.

Line 193: Did you mean environment?

Line 210: Was there an airflow in this paradigm? Or, was it static air?

L346: Please write full "Principal component analysis..."

Line 368: I am excited to see these number (5-9 fold). How would it compare to Culex and Anopheles? Velocities? Distances travelled? In my personal experience, flying Ae. Aegypti to CO₂ was remarkable. Females will orient to CO₂ as neatly as a male moth to its conspecific calling female in a wind tunnel! Any comparative insights/data about the 3 spp will be quite helpful to the scientific community.

Overall, an exciting piece of work from the group. The integrative approaches are phenomenal and highly commendable.

Reviewer #2 (Remarks to the Author):

The manuscript presents the results of a broad series of experiments clarifying how the yellow fever mosquito tracks a host by means of integrating the information gathered by two key sensory channels, olfaction and (color) vision. Furthermore, the results define the type of visual information that governs the approach to objects potentially signaling the presence of a host. In this way, the work depicts fundamental aspects of host tracking and approaching, evidencing that CO₂ is a key factor triggering visually-mediated host tracking that is based both on the color hue and contrast of the object with the background. Astonishingly, this is absolutely absent if increased levels of carbon dioxide transported by flowing air are not detected. The latter feature was initially defined by Van Breugel et al (2015) but mechanistically clarified here.

These facts represent a qualitative leap in our comprehension of host search mechanisms in the main mosquito vectoring arboviruses to humans. Previous knowledge had established many relevant factors mediating host recognition and approach. But this was mostly related to the use of odors and heat, with the role of mosquito vision on host recognition and tracking being poorly understood to date. The current work establishes that vision, in the presence of elevated levels of carbon dioxide simulating those emitted by a host, triggers the recognition of the prevailing color hues of human skin, which are preferred over other visual stimuli of shorter wavelengths or even white objects.

The results support the conclusions and claims, even though authors could be more cautious, especially when they based some of their claims on results obtained using groups of insects. Even though this is mentioned in the methods section, authors could elaborate on this beforehand, as using groups of individuals (and repeated measures) renders these data dependent, and the tests used compromised. This is clearly mentioned in the Methods section and the limited assumption of independence is made clear there. But the reader only gets to this information late in the study after having made a general impression on the meaning of those results. I want to highlight that this reviewer does not consider this as a flaw canceling the evidence, but mentioning limitations when reporting wind tunnel experiments means providing the real message.

Statistical analyses used seem proper in most cases. As mentioned, in some instances they are based on independence assumptions that are not granted. This is the case because trajectories in a trial, for example, represent repeated measures potentially obtained from the same individual. This is true, not to mention that groups of mosquitoes flying together could influence each other, again compromising data independence. This reviewer wants to make clear that the diversity of methodologies used here together with the consistent results presented in most cases make the authors' main claims quite robust. Nevertheless, authors should acknowledge that data obtained with focal individuals in isolation provide more robust facts susceptible to stronger statistical processing and predictive power. Furthermore, this can be overcome in future studies. In my perspective, these issues should not affect the publication of this study but would benefit from a more straightforward narrative acknowledging them as potential limitations during the interpretation of behavioral results obtained in the wind tunnel simultaneously using 50 insects.

The methodology is fundamentally sound, having many instances where authors demonstrate a deep and broad comprehension of the literature and theoretical framework related to the study of sensory biology and behavior in insects. Therefore, the work has higher standards in comparison with most of the current literature in the field. In fact, the flaws listed are quite commonly found in papers that study insect behavior. Overall, I am very positive about the relevance of this study to our comprehension of mosquito host tracking, a fundamental aspect needing deep understanding if the control of these pest insects is expected to become rational in the near future.

Reviewer #3 (Remarks to the Author):

The manuscript submitted by San Alberto and co-workers presents an interesting and original study of the interaction between visual and olfactory cues in mosquitoes. The work is in line with other relevant contributions of the team on the role of visual information and multimodal integration during host searching. The study combines behavioural and physiological measures, establishing a link with molecular studies on the expression and roles of opsins, constituting a significant contribution to our knowledge on mosquito sensory biology.

The evidence presented is convincing as a whole. Still, this reviewer has two concerns. The first one refers to the fact that authors use a terminology centred on the human visual system (e.g. "color", "hue", "illuminance", "white", etc.) and they apply it directly to mosquitoes, which can lead to confusion of non-specialised readers. In a few passages, it is indicated that mosquitoes do not see as humans, but in most of the text, misunderstanding is possible.

Authors are of course experts on the subject and they declare: "...visual stimuli are referred to as they are perceived by the human eye without implying that mosquitoes have the same subjective experience of color...", but in the opinion of this reviewer, they have not been rigorous enough to avoid possible confusion. If we analyse this very sentence, "...have the same subjective experience of color..." implicitly excludes the possibility that mosquitoes could not experience colours at all. In a general way, given that colour vision is a subjective experience which existence in mosquitoes has not been yet demonstrated, it is preferable referring to physical properties of the light exclusively, without any perceptual repercussion. Hue discrimination experiments provide evidence in this sense, but are not definitive.

Follow some examples, needing the attention of the authors:

- The use of the word "color" from the title, without any evidence suggesting that mosquitoes have colour vision does not seem adequate.
- Hue and "color loci" are perceptual attributes, specific to a given species (in this case, humans). The physical analogy to the perceptual attribute "hue" is the "dominant wavelength" or "equivalent wavelength". Color loci and perceptual distances among them can only be determined from perceptual color spaces and are species specific (please, refer to the classical literature on honey-bees).
- "White", as perceived by human vision, does not necessarily mean the even reflection of all wavelengths in physical terms, but the equal stimulation of human photoreceptors. Is it "white" for whose eyes, those of the experimenter or those of the mosquito? It is not indicated in the manuscript, but one can deduce that the grey curve in Fig. 2g corresponds to what authors call "white". I suggest using the expression "evenly reflecting target" or equivalent, to avoid confusion.

My second concern is the statistics treatment because it seems that pseudoreplication has been committed in behavioural experiments, increasing the probability of false significant differences. This problem seems to occur twice. First, when authors state that their experimental approach did not allow obtaining independent data, but they still made the choice of analysing trajectories using statistical methods for which independence is mandatory. Secondly, when they compare "preference indexes", which combine complementary data, i.e., "...the time spent investigating a colored object minus the time investigating the white object, divided by the sum of the times spent investigating the colored and white objects." This kind of index is only useful for illustrating a phenomenon, but employing them for statistical comparisons overestimates differences. The right procedure would be comparing the time spent over an experimental target in discrimination situations vs the time spent over one target in fully control conditions (two identical targets). The

problem is that pseudoreplication biases the results by increasing the Type I error proportionally the size of the data set, which is particularly big in these experiments (please, refer to Fig. 4 in Hulbert S.H., *Ecol. Mon.*, 1984).

Minor points:

- Describing volunteers by race and ethnicity, as defined by administrative procedures of a specific country does not seem very appropriate.
- The reference on kissing-bugs is not entirely right: Reisenman et al. (2000) actually showed that olfactory and visual integration occurs during bugs aggregation, but not that "... kissing bugs prefer visual objects only when also associated with odors." as indicated in the manuscript.
- Fig. 2b. Is the flower species which reflectance is presented particularly preferred by *Aedes aegypti*? Otherwise, flowers can be very different in terms of colour, and this curve loses its interest.
- Fig. 2b. The reflectance of the pond is quite intriguing. It seems that no light at all is reflected by the water surface, which appears as highly counter-intuitive. Please, clarify.
- Fig. 2g. Please, better explain the figure and the axis. The grey curve (corresponding to?) runs beyond 100% on a relative scale. How is this possible? Percentage of what? The different wavelengths below each curve? A particular reference?
- Please, indicate how the Weber contrast was computed. Luminance and illuminance are photometric measures, i.e. adjusted to the spectral sensitivity bell of the human eye. Energy units seem to be more adequate for a non-biased evaluation.
- The ERG in Fig. 4d, along with the study of Giraldo-Calderón (2017, cited) suggests that mosquitoes possess dichromacy, which could make possible the distinction between short and long wavelengths, provided that the proper neural wiring is also present. If the authors agree, what are the consequence for colour discrimination?

As indicated above, this is a relevant contribution and the abovementioned issues can be solved by revising the text and reanalysing some data. I do agree with the authors that in the case of mosquitoes the question of colours relative attractiveness from the human perspective (skin, clothes, traps) is highly relevant, but it is important to be cautious for avoiding confusion, in particular concerning vectors of serious diseases.

REVIEWER COMMENTS AND AUTHOR RESPONSE

Author responses are in blue, Reviewer comments in black

AUTHOR SUMMARY TO MAJOR REVIEWER COMMENTS: The Reviewers had related comments on the potential for statistical bias by the co-release of 50 mosquitoes into the working section and the inability to identify the mosquitoes through the entirety of the experimental trial. These methods and statistical analyses were developed by colleagues and ourselves, the results of which have been published elsewhere, including *Current Biology* and *Nature* (van Breugel, 2014; van Breugel et al., 2015; Vinauger et al., 2018; van Breuge et al., 2018; Zhan et al., 2021). Nonetheless, we have been considering this topic for many years, and we address it by (1) further description of our methods and analyses, (2) additional experiments, and (3) additional multilevel statistical analyses. These points are outlined immediately below.

(1) Our methods and analyses. We provide additional detail in the *Materials and Methods - Trajectory analyses* section on our statistical analyses. Briefly, we use an internal control (e.g., “white” circle) that allows us to compare both between and within (e.g., using a t-test) treatments. Even if there is potential bias or unequal sampling, the internal control provides a method to examine the mosquito preference for the different objects. The bootstrap approach allows us to calculate the confidence intervals and to sample the entire distribution of the data. Nonetheless, we note that a potential for a type-I error still exists even with the bootstrap approach, since there is a potential for a biased sampling (e.g., all data comes from one mosquito in a trial). This brings us to points #2 and #3.

(2) Additional experiments. Details on the additional experiments using individual mosquitoes are now included in the *Materials and Methods - Trajectory analyses* section. To determine the proportion of mosquitoes released into the tunnel that may investigate the visual objects, we performed experiments by individually releasing mosquitoes (n=50) into the tunnel and comparing their behaviors to mosquitoes that were co-released. All 50/50 mosquitoes flew in the tunnel, which compares with the 99.4% of mosquitoes that flew in the tunnel in all the co-released experiments. In addition, we found that flight velocities, durations, and PIs to red and white circles were not statistically different from co-released mosquitoes (unpaired t-tests, $P = 0.75, 0.93, \text{ and } 0.44$ for flight duration, flight velocity, and PI, respectively). During the CO₂ exposure, on average and for those mosquitoes that investigated the visual objects, one mosquito produced 9 flight trajectories during the 1 hr period (range from 1-37 trajectories) and investigated the visual object 3.7 times (± 2.33). This approximated the estimated number of trajectories per mosquito that investigated the objects in co-release experiments (4.76 ± 3.64). Could one mosquito be sampled enough times to account for the number of object investigations we found during the group experiments? Our results suggest “no” because a single mosquito never produced the number of observed investigations to the visual objects in our group experiments.

We also found that 38% of the mosquitoes (19/50) investigated the visual objects. This proportion reflects our reason for co-releasing the mosquitoes: we needed a method to efficiently capture the flight behaviors and responses to the CO₂ and visual objects, and the high number of tested visual stimuli prevented us from individual tests. But this proportion gives us confidence that our trials were not biased from insufficient sampling. This latter point brings us to #3.

(3) Additional analyses. To further explore any differences in the experimental trials and treatment interactions, we performed a multilevel model using the restricted maximum likelihood method. This method provides an unbiased estimate of the variance in the sampling and allows us to examine the effects of the experimental trial (as a random effect) on the results. Using this method, we find that the model results were similar to our previous analyses. The model showed that the experimental trial had no significant impact on the results; instead, the visual treatment was the significant variable ($P = 0.0001079$). We now describe this analysis in the *Materials and Methods* section.

(Reviewer #1)

This manuscript reports interesting findings related to the potential host location strategies in the *Aedes aegypti* mosquitoes. The topic is quite interesting, and authors have utilized an array of clever techniques to address some fascinating questions that have long been puzzling mosquito biologists on what makes *Ae. Aegypti* mosquitoes so well adapted in locating humans. While olfactory biologist and chemists have long celebrated a blend of lactic acid and carbon dioxide (sometimes added with ammonia) as a strong bait in the wind tunnel --

and with decent success in the field studies – the question remained as to how to improvise it. This m/s sheds light onto this topic by elegantly integrating the genetic/molecular techniques with sensory biological paradigms. This work significantly contributes to our fundamental understanding of the olfactory and visual components of the mosquito host seeking behavior esp, in *Aedes*, but also extends broadly to the *Culex* and *Anopheles*. This work once again proves that *A. aegypti* mosquitoes are ‘well-behaved’ organisms to study in the laboratory; authors heroically added a species each of *Culex* and *Anopheles*, and I commend authors for that.

I have a few major concerns about the methodologies that I believe authors should be able to address/explain fairly.

1- Why was it deemed necessary to test mosquitoes in groups (#50)? It appears that the integrity of the tracks cannot be assigned to an individual mosquito (authors state this as much; Line 680). It is very hard for me to conclude that the tracks represent a clear behavioral output from a ‘behaving/responding’ mosquito. Ideally, an individual mosquito’s behavior would have provided a clearer track. I ask authors to explain and expand the reasons for testing mosquitoes in groups, esp. in the context of integrating the tracks. I am certain that the authors must have rigorously thought/debated about it. A clear explanation will be of great help to navigate the work. In the same context, a simple definition of “attraction” (as per the authors) right at the outset will be helpful.

AUTHOR RESPONSE: Please see the above response about the track analysis. Also, we now include a definition for “attraction” on line Page 4, line 88.

2- I am surprised at the integrity of the CO₂ plume. Since CO₂ is heavier than the air, it usually tends to fall down soon after the release from a point source. Authors mention hundreds of measurements to depict the Fig 2d. Can you expand?

AUTHOR RESPONSE: In the figure, we plotted the plume in the x,y plane and not the x,z plane.

3- Was it necessary to have the length of testing for 3hrs, with an hour of CO₂ stimulation? How would you account for habituation, and such effects?

AUTHOR RESPONSE: Under certain bioassay conditions where the flying mosquito is always experiencing the plume (e.g., within a small cage, or a small olfactometer that limits the flight trajectories), the mosquito may adapt or habituate to the plume. By contrast, the working section of the wind tunnel and the size of the CO₂ plume (0.0004 proportional volume of the working section) causes the mosquito to encounter the odor only intermittently (see below figure). We and our colleagues have previously published our characterization of the plume and mosquito and fly encounters with the plume in the 3 to 5 h experimental duration and odor exposure of 1 to 2 h (Van Breugel and Dickinson, 2014; Van Breugel et al., 2015; Vinauger et al., 2018; Van Breugel et al., 2019; see below references). The results of this work showed that this experimental duration had no effect on olfactory and visual responses. In the current study, the 1-hour period was based on this previous work.

To demonstrate the lack of adaptation during the experiment, we included results of the percent of the time that mosquitoes spent investigating the visual objects during the air condition, CO₂ plume, and post-CO₂ plume conditions (see Figures S1e; Fig. 3e). Importantly, during the CO₂ exposure there was no significant difference in the mosquitoes investigating the visual objects at 15 and 45-minute time periods after CO₂ exposure (Kruskal-Wallis test: Chi-sq: 0.73; P = 0.81), thus demonstrating the experimental duration is a reasonable time window to collect and analyze the visual-guided responses.

Another reason for the prolonged-time period (3 h) is that CO₂ causes a long-term effect on the mosquito that lasts ~8 to 10 minutes. We wanted to capture the return to baseline behaviors in our assay, so the post-CO₂ period was 1 hr.

Reprinted from
 Current Biology,
 24(3), van Breugel,
 F. and Dickinson,
 M.H., Plume-
 Tracking Behavior
 of Flying
 Drosophila
 Emerges from a
 Set of Distinct
 Sensory-Motor
 Reflexes,
 pp.274-286.,
 Copyright 2014,
 with permission
 from Elsevier

*From: van Breugel, F. and Dickinson, M.H., 2014. Plume-tracking behavior of flying Drosophila emerges from a set of distinct sensory-motor reflexes. *Current Biology*, 24(3), pp.274-286.

*See also:

- Zhan, Y., Alonso San Alberto, D., Rusch, C., Riffell, J.A. and Montell, C., 2021. Elimination of vision-guided target attraction in *Aedes aegypti* using CRISPR. *Current Biology*.
- van Breugel, F., Riffell, J., Fairhall, A. and Dickinson, M.H., 2015. Mosquitoes use vision to associate odor plumes with thermal targets. *Current Biology*, 25(16), pp.2123-2129.
- Vinauger, C., van Breugel, F., Locke, L.T., Tobin, K.K., Dickinson, M.H., Fairhall, A.L., Akbari, O.S. and Riffell, J.A., 2019. Visual-olfactory integration in the human disease vector mosquito *Aedes aegypti*. *Current Biology*, 29(15), pp.2509-2516.
- van Breugel, F., Huda, A. and Dickinson, M.H., 2018. Distinct activity-gated pathways mediate attraction and aversion to CO₂ in *Drosophila*. *Nature*, 564(7736), pp.420-424.

Here are some specific comments:

Line 2 and onwards: Use of term 'mosquito' is very loose. Culicine and Anopheline mosquitoes have rather different and distinct biology, including sensory. It would be helpful if authors do not generalize.

AUTHOR RESPONSE: Beginning on page 2, we now specify the species, particularly in the Results section to distinguish between our experiments with *Ae. aegypti* versus those with *Cx quinquefasciatus* and *An. stephensi*.

Line 30. Reference 13 (Lacey and Carde's work) was done with *Culex*

AUTHOR RESPONSE: We have removed this reference for clarity.

Line 56. Consider replacing "... when experiencing..." with "while encountering..."

AUTHOR RESPONSE: Done. We thank the reviewer for the suggestion.

Line 65: What does it signify to say 450? Please explain.

AUTHOR RESPONSE: This 450 body lengths of the mosquito represents the length of the tunnel (human equivalent to 7.5 football fields). This was our attempt to demonstrate the relative size of the tunnel working section.

Line 76: Are you suggesting ~1500 ppm is what a mosquito will experience at 30cm distance from a breathing human? (Considering we exhale ~4-5%)

AUTHOR RESPONSE: No, there is a misunderstanding here. The 1500 ppm was the measurement in the wind tunnel working section. We now clarified this in the manuscript text (Page 4).

Line 81. How would you arrive with that number? (double # getting activated)

AUTHOR RESPONSE: On Fig. 1h, the "flight activity" is relative to the number of flying *Ae. aegypti* mosquitoes in the Air treatment (#CO₂/#Air). This is described in the Figure legend. We now include this information in the **Materials and Methods - Trajectory analyses** section (Page 25).

Line 82. Velocity would assume an individual's distance. Can you please provide the basis for your assumption?

AUTHOR RESPONSE: We believe that the reviewer is asking about the odor-tracking behavior. During CO₂, the mosquitoes spend much of their time within the portion of the tunnel where the plume resides. and the trajectory velocities increase. Moreover, during CO₂ they begin to orient upwind and towards the odor source. By contrast, during Air- or post-CO₂ phases, they exhibit behaviors analogous to moth “casting” where they often orient crosswind.

Line 89: Fig 1Se is not clear. Can you instead just indicate pre-, CO₂ and post-CO₂? (instead of scale 0-200mins). Also in the figure S1 legend, Line 9 from top, add “significantly INCREASES...”

AUTHOR RESPONSE: In Fig 1Se, we now show where the pre-, CO₂, and post-CO₂ intervals occur in the plot, and have clarified this in the legend.

For Figure 1b and throughout: Y axis needs to read “RefLectance”

AUTHOR RESPONSE: Done. We thank the reviewer for catching that typographic error.

Line 105: It may be interesting to have a measure of distance. If certain stimulus (i) induce more kinetic effects.

AUTHOR RESPONSE: We now include a measure of distance in the figure. There were no statistically significant differences in the flight distances between stimuli (i.e., Fig. S1b,d).

Line 157. Legends have small letters whereas figures have CAPS.

AUTHOR RESPONSE: Now corrected!

Line 193: Did you mean environment?

AUTHOR RESPONSE: Yes. Now corrected.

Line 210: Was there an airflow in this paradigm? Or, was it static air?

AUTHOR RESPONSE: There was very little airflow in this assay. There was some airflow from the vents and the CO₂ pad. We now include this information in the Materials and Methods - Visual Attraction to Skin in a Cage-Assay (Page 25). We also include more information on this bioassay on Page 23.

L346: Please write full “Principal component analysis...”

AUTHOR RESPONSE: Done.

Line 368: I am excited to see these number (5-9 fold). How would it compare to Culex and Anopheles? Velocities? Distances travelled? In my personal experience, flying Ae. Aegypti to CO₂ was remarkable. Females will orient to CO₂ as neatly as a male moth to its conspecific calling female in a wind tunnel! Any comparative insights/data about the 3 spp will be quite helpful to the scientific community.

AUTHOR RESPONSE: We are continuing this research, and hope to have a manuscript in the near future that describes the odor-tracking and visual object for the different species, and including An. coluzzii. What was also interesting to us, was that at higher light intensities (>10 lux) the An. stephensi and Cx. quinquefasciatus mosquitoes would respond to CO₂, but would not be attracted to the visual objects. However, at lower light intensities the odor would gate their attraction to visual objects. So the light environment had a strong effect on visual attraction, but not odor-evoked responses, for these nocturnal/crepuscular mosquitoes. We now describe this information in the Results and Methods sections.

Overall, an exciting piece of work from the group. The integrative approaches are phenomenal and highly commendable.

AUTHOR RESPONSE: We thank the reviewer for the kind comments!

(Reviewer #2)

The manuscript presents the results of a broad series of experiments clarifying how the yellow fever mosquito tracks a host by means of integrating the information gathered by two key sensory channels, olfaction and (color) vision. Furthermore, the results define the type of visual information that governs the approach to objects potentially signaling the presence of a host. In this way, the work depicts fundamental aspects of host tracking

and approaching, evidencing that CO₂ is a key factor triggering visually-mediated host tracking that is based both on the color hue and contrast of the object with the background. Astonishingly, this is absolutely absent if increased levels of carbon dioxide transported by flowing air are not detected. The latter feature was initially defined by Van Breugel et al (2015) but mechanistically clarified here.

These facts represent a qualitative leap in our comprehension of host search mechanisms in the main mosquito vectoring arboviruses to humans. Previous knowledge had established many relevant factors mediating host recognition and approach. But this was mostly related to the use of odors and heat, with the role of mosquito vision on host recognition and tracking being poorly understood to date. The current work establishes that vision, in the presence of elevated levels of carbon dioxide simulating those emitted by a host, triggers the recognition of the prevailing color hues of human skin, which are preferred over other visual stimuli of shorter wavelengths or even white objects.

The results support the conclusions and claims, even though authors could be more cautious, especially when they based some of their claims on results obtained using groups of insects. Even though this is mentioned in the methods section, authors could elaborate on this beforehand, as using groups of individuals (and repeated measures) renders these data dependent, and the tests used compromised. This is clearly mentioned in the Methods section and the limited assumption of independence is made clear there. But the reader only gets to this information late in the study after having made a general impression on the meaning of those results. I want to highlight that this reviewer does not consider this as a flaw canceling the evidence, but mentioning limitations when reporting wind tunnel experiments means providing the real message.

AUTHOR RESPONSE: The reviewer brings up an important point. Please see our response above in the "Author Summary to Reviewer Comments" section. Regarding the reviewer's comment about the group data, we mention it right away in the Results section (Page 4, Line 71: "In each experimental trial, 50 mated *Ae. aegypti* females were co-released into the tunnel..."). We now add more detail in the Results section (on Page 4, starting on Line 73) to better describe the assay and reasoning behind releasing multiple mosquitoes.

Statistical analyses used seem proper in most cases. As mentioned, in some instances they are based on independence assumptions that are not granted. This is the case because trajectories in a trial, for example, represent repeated measures potentially obtained from the same individual. This is true, not to mention that groups of mosquitoes flying together could influence each other, again compromising data independence. This reviewer wants to make clear that the diversity of methodologies used here together with the consistent results presented in most cases make the authors' main claims quite robust. Nevertheless, authors should acknowledge that data obtained with focal individuals in isolation provide more robust facts susceptible to stronger statistical processing and predictive power. Furthermore, this can be overcome in future studies. In my perspective, these issues should not affect the publication of this study but would benefit from a more straightforward narrative acknowledging them as potential limitations during the interpretation of behavioral results obtained in the wind tunnel simultaneously using 50 insects.

AUTHOR RESPONSE: We now include information in the Materials and Methods section on this topic. And as described above in the "Author Summary to Reviewer Comments" section, the co-release of the 50 mosquitoes provided an efficient method to examine the olfactory-visual responses.

A brief note on the topic of in-flight interactions: Based on the flight kinematics and behaviors, we did not observe any differences between when mosquitoes were singly- *versus* simultaneously released. We attribute this because we release only mated females (co-release of males would cause interactions). In addition, compared to our past experiments (Van Breugel et al., 2014; Van Breugel et al., 2015), we lowered the number of mosquitoes per m³ in the working section from 140 to 53 mosquitoes/m³ to further minimize in-flight interactions. We now include additional information in the *Materials and Methods - Trajectory analysis* section (Page 25).

The methodology is fundamentally sound, having many instances where authors demonstrate a deep and broad comprehension of the literature and theoretical framework related to the study of sensory biology and behavior in insects. Therefore, the work has higher standards in comparison with most of the current literature in the field. In fact, the flaws listed are quite commonly found in papers that study insect behavior. Overall, I am very positive about the relevance of this study to our comprehension of mosquito host tracking, a fundamental aspect needing deep understanding if the control of these pest insects is expected to become rational in the near future.

AUTHOR RESPONSE: We thank the reviewer for the kind comments!

(Reviewer #3)

The manuscript submitted by San Alberto and co-workers presents an interesting and original study of the interaction between visual and olfactory cues in mosquitoes. The work is in line with other relevant contributions of the team on the role of visual information and multimodal integration during host searching. The study combines behavioural and physiological measures, establishing a link with molecular studies on the expression and roles of opsins, constituting a significant contribution to our knowledge on mosquito sensory biology. The evidence presented is convincing as a whole. Still, this reviewer has two concerns. The first one refers to the fact that authors use a terminology centred on the human visual system (e.g. "color", "hue", "illuminance", "white", etc.) and they apply it directly to mosquitoes, which can lead to confusion of non-specialised readers. In a few passages, it is indicated that mosquitoes do not see as humans, but in most of the text, misunderstanding is possible.

Authors are of course experts on the subject and they declare: "...visual stimuli are referred to as they are perceived by the human eye without implying that mosquitoes have the same subjective experience of color...", but in the opinion of this reviewer, they have not been rigorous enough to avoid possible confusion. If we analyse this very sentence, "...have the same subjective experience of color..." implicitly excludes the possibility that mosquitoes could not experience colours at all. In a general way, given that colour vision is a subjective experience which existence in mosquitoes has not been yet demonstrated, it is preferable referring to physical properties of the light exclusively, without any perceptual repercussion. Hue discrimination experiments provide evidence in this sense, but are not definitive.

Follow some examples, needing the attention of the authors:

- The use of the word "color" from the title, without any evidence suggesting that mosquitoes have colour vision does not seem adequate.

AUTHOR RESPONSE: We thank the reviewer for the suggestions. The reviewer brings up several important points. Per the reviewer's suggestions, we have changed the manuscript title and removed "color".

- Hue and "color loci" are perceptual attributes, specific to a given species (in this case, humans). The physical analogy to the perceptual attribute "hue" is the "dominant wavelength" or "equivalent wavelength". Color loci and perceptual distances among them can only be determined from perceptual color spaces and are species specific (please, refer to the classical literature on honey-bees).

AUTHOR RESPONSE: We have worked to change the terms of "color", "hue", etc... and now describe the objects based on their peak wavelength. We have also removed "hue" and "color loci", and replaced the terms with "dominant wavelength", or similar terms.

- "White", as perceived by human vision, does not necessarily mean the even reflection of all wavelengths in physical terms, but the equal stimulation of human photoreceptors. Is it "white" for whose eyes, those of the experimenter or those of the mosquito? It is not indicated in the manuscript, but one can deduce that the grey curve in Fig. 2g corresponds to what authors call "white". I suggest using the expression "evenly reflecting target" or equivalent, to avoid confusion.

AUTHOR RESPONSE: We have replaced "white" with "evenly reflecting object".

My second concern is the statistics treatment because it seems that pseudoreplication has been committed in behavioural experiments, increasing the probability of false significant differences.

This problem seems to occur twice. First, when authors state that their experimental approach did not allow obtaining independent data, but they still made the choice of analysing trajectories using statistical methods for which independence is mandatory. Secondly, when they compare "preference indexes", which combine complementary data, i.e., "...the time spent investigating a colored object minus the time investigating the white object, divided by the sum of the times spent investigating the colored and white objects." This kind of index is only useful for illustrating a phenomenon, but employing them for statistical comparisons overestimates differences. The right procedure would be comparing the time spent over an experimental target in discrimination situations vs the time spent over one target in fully control conditions (two identical targets). The problem is that pseudoreplication biases the results by increasing the Type I error proportionally the size of the data set, which is particularly big in these experiments (please, refer to Fig. 4 in Hulbert S.H., Ecol. Mon., 1984).

AUTHOR RESPONSE: The reviewer brings up several important points that we attempt to address in the "Author Summary to Reviewer Comments" section. In each experimental trial, we have an internal control by using the evenly reflecting object ("white") relative to the other object. This provides both an internal control, but also allows us to compare between other treatments, including the

treatment where we used two evenly reflecting objects. We now describe additional analyses that demonstrate that Type I error did not bias our results. Last, we analyze not only the Preference Indices, but also the time spent near the objects, the flight velocities, and the flight durations. We believe that these measures, and others detailed in the manuscript, provide a rigorous analysis of the behaviors to the olfactory and visual cues.

Minor points:

- Describing volunteers by race and ethnicity, as defined by administrative procedures of a specific country does not seem very appropriate.

AUTHOR RESPONSE: Done, now modified.

- The reference on kissing-bugs is not entirely right: Reisenman et al. (2000) actually showed that olfactory and visual integration occurs during bugs aggregation, but not that "... kissing bugs prefer visual objects only when also associated with odors." as indicated in the manuscript.

AUTHOR RESPONSE: Now clarified in the text (Page 19, Line 543).

- Fig. 2b. Is the flower species which reflectance is presented particularly preferred by *Aedes aegypti*? Otherwise, flowers can be very different in terms of colour, and this curve loses its interest.

AUTHOR RESPONSE: Yes, the *Platanthera obtusata* flower is an important nectar source for *Aedes* sp. mosquitoes (Lahondère, C., Vinauger, C., Okubo, R.P., Wolff, G.H., Chan, J.K., Akbari, O.S. and Riffell, J.A., 2020. The olfactory basis of orchid pollination by mosquitoes. *Proceedings of the National Academy of Sciences*, 117(1), pp.708-716.). This information is included in the legend.

- Fig. 2b. The reflectance of the pond is quite intriguing. It seems that no light at all is reflected by the water surface, which appears as highly counter-intuitive. Please, clarify.

AUTHOR RESPONSE: This is an excellent point. The measurements were made within the puddle containing *Aedes* larvae, not on the surface of the puddle. We have clarified this point in the figure legend.

- Fig. 2g. Please, better explain the figure and the axis. The grey curve (corresponding to?) runs beyond 100% on a relative scale. How is this possible? Percentage of what? The different wavelengths below each curve? A particular reference?

AUTHOR RESPONSE: The traces in Fig. 1g are the Reflectance of the visual stimuli used in the experiments, as quantified using a spectrophotometer and calibrated with a white Spectralon standard. Different traces correspond to the different visual stimuli. This information is now clarified in the figure legend and figure.

- Please, indicate how the Weber contrast was computed. Luminance and illuminance are photometric measures, i.e. adjusted to the spectral sensitivity bell of the human eye. Energy units seem to be more adequate for a non-biased evaluation.

AUTHOR RESPONSE: The reviewer brings up an important point. Yes, we did use energy as the measured variable. The Weber contrasts were calculated as described in the Materials and Methods section (Page 22, Line 618) as: "The achromatic contrasts of the visual objects relative to the background were measured using the calibrated spectrophotometer and the Weber contrasts calculated by the intensity ($\mu\text{W}/\text{cm}^2$) of the object (I_{object}) and background ($I_{\text{background}}$), where $(I_{\text{object}} - I_{\text{background}})/I_{\text{background}}$."

- The ERG in Fig. 4d, along with the study of Giraldo-Calderón (2017, cited) suggests that mosquitoes possess dichromacy, which could make possible the distinction between short and long wavelengths, provided that the proper neural wiring is also present. If the authors agree, what are the consequence for colour discrimination?

AUTHOR RESPONSE: We agree with the reviewer. In the manuscript Discussion, we describe the possibility for color opponent responses. In the case of dichromacy, color opponency between the S and M/L wavelength photoreceptors is favored, especially when the differences in the color space become one-dimensional.

REVIEWERS' COMMENTS

Reviewer #1 (Remarks to the Author):

The revised m/s addressed all my major concerns; and I am delighted to see the comparison of individual mosquito behavior with the 'group'.

Zain Syed

Reviewer #3 (Remarks to the Author):

My congratulations to the authors for their contribution to our understanding of mosquito sensory biology.